# DDRL: A Diffusion-Driven Reinforcement Learning Approach for Enhanced TSP Solutions

## Abstract

The Traveling Salesman Problem (TSP) is a fundamental challenge in combinatorial optimization, known for its NP-hard complexity. Reinforcement Learning (RL) has proven to be effective in managing larger and more complex TSP instances, yet it encounters challenges such as training instability and necessity for a substantial amount of training resources. Diffusion models, known for iteratively refining noisy inputs to generate high-quality solutions, offer scalability and exploration capabilities for TSP but may struggle with optimality in complex cases and require large, resource-intensive training datasets. To address these limitations, we propose DDRL (Diffusion-Driven Reinforcement Learning), which integrates diffusion models with RL. DDRL employs a latent vector to generate an adjacency matrix, merging image and graph learning within a unified RL framework. By utilizing a pre-trained diffusion model as a prior, DDRL exhibits strong scalability and enhanced convergence stability. We also provide theoretical analysis that training DDRL aligns with the diffusion policy gradient in the process of solving the TSP, demonstrating its effectiveness. Additionally, we introduce novel constraint datasets—obstacle, path, and cluster constraints—to evaluate DDRL's generalization capabilities. We demonstrate that DDRL offers a robust solution that outperforms existing methods in both basic and constrained TSP problems. The code used for our experiments is available anonymously for review[1].

## 1 Introduction

The Traveling Salesman Problem (TSP) is a classical problem in combinatorial optimization and theoretical computer science. Given a set of cities and a distance function that determines the distance between each pair of cities, the objective is to find an order in which to visit these cities that minimizes the total tour length. Despite its straightforward definition, the TSP is renowned for its computational complexity, classified as NP-hard, which has led to extensive research in developing algorithms and optimization methods (Cheikhrouhou & Khoufi, 2021).

In recent years, machine learning techniques, including deep neural networks, have gained attention for addressing complex optimization problems like the TSP. Reinforcement Learning (RL) has shown promise in solving sequential decision-making tasks Sutton & Barto (2018), with efforts to combine RL with models like Graph Neural Networks (GNNs) (Kool et al., 2018) and Transformers (Bresson & Laurent, 2021) to enhance performance. However, these approaches have limited effectiveness in handling larger and more complex TSP instances, particularly in terms of autoregressive decoding and generalization. Additionally, RL models often suffer from instability, requiring extensive training to achieve optimal solutions (Bresson & Laurent, 2021), highlighting the need for further refinement to tackle more challenging TSP cases.

Diffusion models, a type of generative models, iteratively refine noisy inputs to produce high-quality solutions (Ho et al., 2020). These models are effective in exploring diverse solution spaces, avoiding local minima, and consistently generating near-optimal solutions across TSP instances of varying sizes. Their iterative process provides scalability and robustness. However, diffusion models have

---

[1]Anonymous code repository: `https://anonymous.4open.science/r/diffusion_rl_tsp`

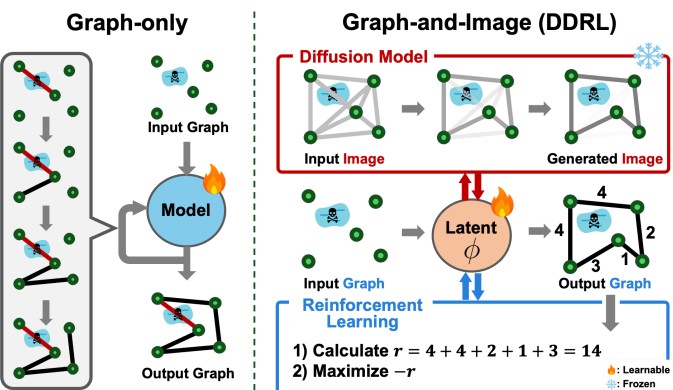

Figure 1: Comparison of traditional TSP methods and our proposed DDRL framework. Traditional methods (left) face two main limitations: they rely on autoregressive inference, which leads to increasing computational costs as the number of cities grows, and they struggle to effectively handle constraints, such as avoiding restricted zones shown in the figure. In contrast, DDRL (right) generates solutions at the image level, making inferences independent of the number of cities, and effectively handles constraints by leveraging visual features. By integrating both the graph and image domains, DDRL enhances both efficiency and solution quality.

limitations—they may struggle to find optimal solutions in complex cases, rely on large training datasets, and exhibit reduced performance when applied to TSP instances that differ significantly from the training data.

To address the limitations of previous approaches, we propose a new framework called Diffusion-Driven Reinforcement Learning (DDRL), which leverages the complementary relationship between images and graphs to solve the TSP. DDRL reinterprets the problem in the image domain by integrating graph structure-based RL with the connection between the Markov Decision Process (MDP) and the diffusion process. This approach reduces vulnerability to scalability issues from increasing instance sizes, as it maps the problem to an image space independent of the number of nodes. Furthermore, we utilize pre-trained diffusion models on image data as prior knowledge, significantly enhancing the convergence stability of the learning process. The integration of RL with diffusion models further improves resource efficiency, reduces data dependency, and increases scalability and robustness, enabling more efficient and adaptive solutions to complex optimization problems. As shown in Figure 1, traditional graph-only approaches rely on autoregressive inference, leading to higher computational costs as the number of cities increases and difficulty in handling constraints such as restricted zones. In contrast, DDRL integrates both graph and image levels, making it independent of the number of cities while effectively addressing constraints through the use of visual features. This combined approach leads to enhanced efficiency and solution quality.

We validate the effectiveness of our proposed methodology through extensive experiments, comparing it with state-of-the-art baseline models. We assess its scalability across a diverse range of instances, from small sets to large-scale problems. In addition, we evaluate its performance on three hand-conditioned visually evident constraint datasets (Obstacle, Path, and Cluster) featuring novel constraints. These datasets, which are visually intuitive and simple, are very challenging for conventional methods. The results indicate that DDRL not only provides more accurate solutions but also achieves more efficient and robust learning than existing approaches.

The main contributions of this research can be summarized as follows:

- To the best of our knowledge, we first integrate a diffusion model into the RL approach, leveraging visual capabilities to solve the TSP problem.

- We demonstrate the theoretical basis of DDRL and its robustness and scalability across a range of problem sizes.

- We introduce novel, visually intuitive constraint datasets, showing that DDRL outperforms a wide range of TSP algorithms, excelling both in standard TSP settings and in handling complex constraint conditions.

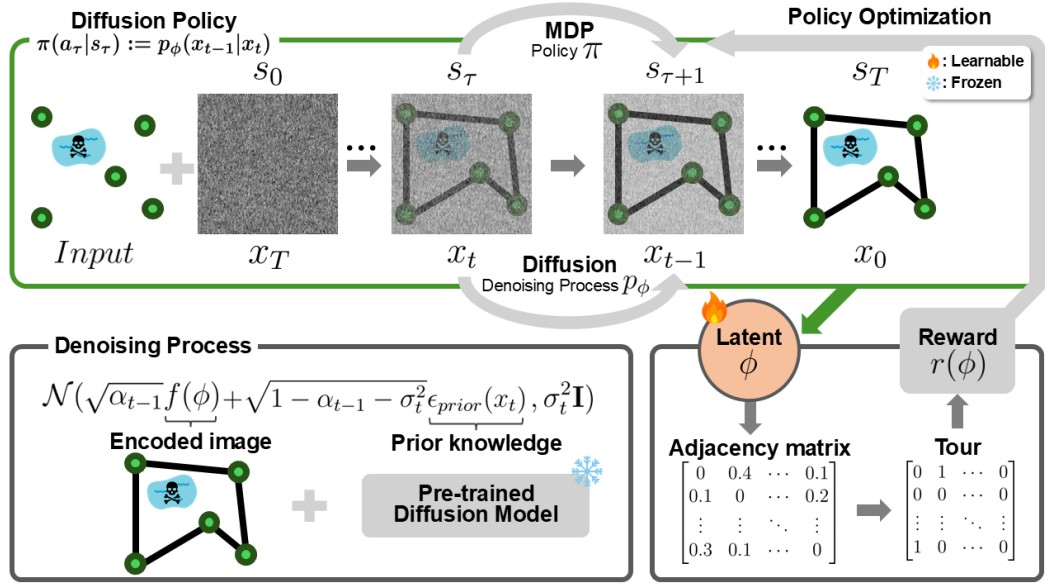

Figure 2: Overall process of the proposed method integrating diffusion models with RL for solving the TSP. It illustrates the integration of a learned latent vector, prior knowledge from a pre-trained diffusion model, and the RL framework to achieve high-quality TSP solutions. The process starts with random noise combined with location information to form the initial latent image $x_T$. This image is iteratively refined through a sequence of denoising steps, guided by the pre-trained diffusion model $\epsilon_{prior}$, until it transforms into the final generated image $x_0$. A reward function, based on the total tour length derived from the generated image, is then used to optimize the latent vector $\phi$ over multiple epochs.

## 2 RELATED WORK

### 2.1 TSP

The TSP involves finding the shortest route that visits a set of cities exactly once before returning to the start. Exact methods, such as Dynamic Programming (Held & Karp, 1962) and Integer Programming, guarantee optimal solutions but become infeasible for large instances. Heuristic approaches, including the Christofides Algorithm (Christofides, 2022), 2-Opt (Lin, 1965), and LKH-3 (Helsgaun, 2017), trade optimality for efficiency but still require significant computational resources as the problem size increases. In other words, heuristic methods face performance limitations and become excessively slow as the number of cities, $N$, increases. On the other hand, our DDRL reduces training time and enhances performance by integrating graph and visual information, making it more scalable to larger instances of TSP.

### 2.2 REINFORCEMENT LEARNING

Reinforcement Learning (RL) (Sutton & Barto, 2018) is a framework for solving sequential decision-making problems through the optimization of a policy using MDPs (Puterman, 1990). Recent advancements in RL have improved its application to TSP by incorporating models like Graph Neural Networks (GNNs) (Kool et al., 2018) and Transformers Bresson & Laurent (2021). However, RL faces challenges such as instability during training and the need for substantial training resources (Bresson & Laurent, 2021). DDRL addresses these challenges by stabilizing convergence and enhancing efficiency through the use of pre-trained diffusion models as prior knowledge.

### 2.3 DIFFUSION MODELS

Diffusion models, such as Denoising Diffusion Probabilistic Models (DDPM) (Ho et al., 2020) and Denoising Diffusion Implicit Models (DDIM) (Song et al., 2020), excel at generating high-quality

outputs by iteratively refining noisy inputs. Applied to combinatorial optimization problems like TSP, diffusion models improve solution quality by leveraging iterative refinement processes (Sun & Yang, 2024), although they often require extensive labeled datasets and face trade-offs between efficiency and solution accuracy. In this work, we integrate RL with pre-trained diffusion models, like Stable Diffusion (Rombach et al., 2022), to improve stability and generalization in TSP solutions while reducing reliance on large training datasets.

## 3 METHOD

In this section, we present a methodology based on RL and diffusion models to solve the TSP. Section 3.1 outlines the problem definition as a multi-step MDP. Section 3.2 details the structure and function of the policy network. Finally, Section 3.3 describes the DDRL optimization process. The entire workflow is visualized in Figure 2.

### 3.1 MULTI-STEP MDP FOR SOLVING TSP PROBLEM

The denoising diffusion process is reformulated as a multi-step MDP in RL (Black et al., 2023; Zhang et al., 2024). The following definitions clarify the connection between these two distinct approaches. As the time step $\tau$ in the MDP increases, the denoising step $t$ decreases, where $\tau$ and $t$ are related by the equation $\tau = T - t$. The crucial elements for defining an MDP are the state, action, and policy, specified as follows:

$$s_\tau := (t, x_t)$$
$$a_\tau := x_{t-1}$$
$$\pi(a_\tau|s_\tau) := p_\phi(x_{t-1}|x_t).$$

Here, the action in RL at time step $\tau$ is defined as the one-step denoised image $x_{t-1}$ in the diffusion process. This definition allows the policy $\pi$ in the MDP to represent the probability of a denoising step, parameterized by $\phi \in \mathbb{R}^{N \times N}$. The matrix $\phi$ is a latent vector normalized (non-diagonal, symmetric, etc.) to obtain an adjacency matrix. This adoption of $\phi$ enables the estimation of paths and computation of TSP rewards in terms of the graph structure. The sampling process, which generates the sampled data $x_0$ from pure noise $x_T$, consists of a sequence of denoising steps. Namely, the policy at time step $\tau$ corresponds to the generative process at denoising step $t$, transforming the image $x_t$ into the one-step refined image $x_{t-1}$. Inspired by DDIM (Song et al., 2020), which introduces a non-Markovian framework conditioned on $x_0$, we adopt this non-Markovian setting and estimate $x_0$ using $f(\phi)$. This conditioning modifies the denoising function as follows:

$$p_\phi(x_{t-1}|x_t) := \begin{cases} \mathcal{N}(f(\phi), \sigma_t^2 \mathbf{I}) & \text{if } t = 1 \\ q_\sigma(x_{t-1}|x_t, f(\phi)) & \text{otherwise} \end{cases}$$

$$\text{where } q_\sigma(x_{t-1}|x_t, f(\phi)) = \mathcal{N}(\mu(x_t, \phi), \sigma_t^2 \mathbf{I}). \tag{1}$$

In this setting, $\sigma_t^2$ denotes the variance of the noise at time step $t$. The function $f$, in the same manner as Graikos et al. (2022) encodes image, deterministically maps a latent vector $\phi$ to a refined image which also serves as an estimate of $x_0$. This approach is valid for two reasons: First, the upscaling process preserves the probability distribution of the adjacency matrix, effectively translating the graph structure into an image domain. Second, the mapping of this image-form probability distribution corresponds to the most probable adjacency matrix state, thus predicting $x_0$.

$$\mu(x_t, \phi) = \sqrt{\alpha_{t-1}} f(\phi) + \sqrt{1 - \alpha_{t-1} - \sigma_t^2} \epsilon_{prior}(x_t),$$

where $\alpha_t$ is a noise schedule parameter at time step $t$ that controls the amount of noise added or removed. $\epsilon_{prior}$ is a pre-trained diffusion model constructed in a deterministic way. The final denoising process is as follows:

$$x_{t-1} = \sqrt{\alpha_{t-1}} f(\phi) + \sqrt{1 - \alpha_{t-1} - \sigma_t^2} \epsilon_{prior}(x_t) + \sigma_t \epsilon, \tag{2}$$

where $\epsilon$ is sampled from $\mathcal{N}(\mathbf{0}, \mathbf{I})$ and the pre-trained diffusion model $\epsilon_{prior}$ (referred to as *prior knowledge*) guides the denoising process. This prior knowledge, trained on TSP labeled data, transforms a noisy image $x_T$ with random city connections into an image $x_0$ with optimal connections.

Unlike typical fine-tuning, the prior knowledge remains fixed during RL training, providing consistent directional guidance for path connections.

Given the inherent complexity of combinatorial optimization problems, such as the TSP, utilizing a multi-step MDP framework that decomposes the problem into smaller sub-tasks is more advantageous than attempting to solve it with a single action in a one-step MDP. Accordingly, this paper employs the multi-step MDP approach as its primary methodology.

### 3.2 Policy Gradient as a Denoising Process

Designing an appropriate reward to solve combinatorial problems in image domains presents a significant challenge, as detecting paths within the denoising process is demanding. To overcome this, we introduce the latent vector $\phi$ at the graph level, where edge values are more manageable. A reward signal $r(\phi)$, defined at time step $\tau = T - 1$, is calculated as the negative total length of the tour derived from $\phi$.

Based on this calculation, the reward $R(s_\tau, a_\tau)$ is defined as the reward signal $r$ when the time step $\tau = T - 1$ in RL nearly reaches the final time step, while the diffusion step $t$ approaches one as follows:

$$R(s_\tau, a_\tau) := \begin{cases} r(\phi) & \text{if } \tau = T - 1 \\ 0 & \text{otherwise.} \end{cases} \tag{3}$$

The definition is straightforward, as the tour is calculated at the stage when the image has been refined through $T$ times. The return in RL, which is the sum of future rewards in a simple way, precisely corresponds to $r(\phi)$. This correspondence implies that solving the multi-step MDP is equivalent to maximizing an objective function. To train the diffusion model within the RL framework, we define the objective function based on the diffusion model's sampling process. Using the sampling distribution $p_\phi(x_t)$, we set the RL objective function to maximize the reward signal $r$, which is based on the sample $x_t$ as follows:

$$J(\phi) = \mathbb{E}_{x_t \sim p_\phi(x_t)}[r(\phi)]. \tag{4}$$

This loss function also addresses the problem of constrained TSP, where $r(\phi)$ is subject to certain constraints, such as requiring some elements of $\phi$ to be zero. It is essential to ensure that the objective function, derived from our custom-designed reward with $\phi$, aligns with the direction of learning in the diffusion policy. This alignment allows our loss gradient to be calculated as the diffusion policy gradient, with the policy defined by the diffusion-generating process. The following proposition, based on the researches (Fan & Lee, 2023; Black et al., 2023), demonstrates this alignment.

**Proposition 1** *The gradient of our objective function defined in Equation 4 is equivalent to a diffusion policy gradient update:* $\nabla_\phi J(\phi) = \mathbb{E}_{s_{0:T}} \left[ r(\phi) \sum_{\tau=0}^{T-1} \nabla_\phi \log \pi(a_\tau | s_\tau) \right]$.

The theoretical equivalence established by the proposition allows effective learning, even when different techniques are applied. Specifically, the proposition ensures that the direction of the diffusion policy gradient aligns with the goal of policy gradient learning. The policy is optimized to reduce the loss function, which is parameterized by $\phi$. Our method operates at the image level through diffusion denoising while implicitly learning at the graph level by optimizing latent vectors, which also serve as adjacency matrices. This approach reduces the number of parameters, simplifies the learning process, and offers tailored solutions for each problem. By calculating the tour length directly from the learned adjacency matrix $\phi$, we can compute rewards without the complications of blur or noise often encountered in image-level methods. Integrating diffusion model-based learning with an RL framework, our approach delivers superior results for TSP compared to existing methods. The detailed derivation of the proposition and its theoretical foundation are provided in the appendix.

### 3.3 DDRL Optimization

In this section, we outline the overall process of DDRL, focusing on how the combination of diffusion models and RL optimizes solutions for the TSP. The core of our method is the Diffusion-Driven RL optimization, which consists of two key phases: the *sampling phase*, where tours are generated and evaluated, and the *policy update phase*, where the latent vector $\phi$ is refined using gradient-based

optimization. The process starts with initializing the latent vector $\phi$, which is done by minimizing the diffusion loss using prior knowledge $\epsilon_{prior}$. This initialization step enhances the stability of the optimization process and reduces variance by leveraging pre-trained diffusion models to provide a reliable starting point.

**Sampling Phase:** During this phase, DDRL generates tours by iteratively sampling latent vectors $\phi_i$ and applying the diffusion process. The sequence of images $\{x_T, x_{T-1}, \ldots, x_0\}$ is produced following the denoising process described in Equation 2, where $\phi$ guides the refinement of noisy inputs. Each final image $x_0$ represents a Hamiltonian graph corresponding to a potential solution for the TSP. The solution is expressed as a tour, denoted by $T_i$, a list of city indices representing the order in which the cities are visited. A 2-opt local search is applied to improve the generated tour $T_i$ and further optimize the total tour length while adhering to any problem-specific constraints. After each tour $T_i$ is computed, the reward $r(\phi_i)$, based on the negative tour length, is calculated. The advantage $A_i = r_i - \bar{r}$, where $\bar{r}$ is the mean reward across all samples, helps quantify the quality of the sampled tours.

**Initialization (diffusion loss minimization)** Proper initialization is crucial for the convergence of our algorithm. We employ a specialized method using prior knowledge $\epsilon_{prior}$ and diffusion loss to enhance initial stability. Initialize the latent vector $\phi$ by minimizing the diffusion loss gradient:

$$\phi \leftarrow \phi - \lambda \nabla_\phi \left| \epsilon - \epsilon_{prior} \left( \sqrt{\overline{\alpha_{t_i}}} f(\phi) + \sqrt{1 - \overline{\alpha_{t_i}}} \epsilon, t_i \right) \right|_2^2, \tag{5}$$

where $\overline{\alpha_{t_i}}$ is a modified cumulative product over time steps $t_i$, $\epsilon$ represents the noise in the diffusion process, and $\lambda$ is the learning rate. $\phi$ is optimized for each timestep $t$ using the gradient in Equation 5. DDRL performs multiple initialization attempts to secure the best possible latent vector for the given TSP instance.

**Policy Update Phase:** After calculating the rewards, the policy parameters $\phi$ are updated using gradient ascent:

$$\phi \leftarrow \phi + \alpha \nabla_\phi J(\phi), \tag{6}$$

where $J(\phi)$ denotes the expected reward and $\alpha$ is the learning rate. This update step refines the policy by adjusting the latent vector $\phi$ to maximize the reward, leading to progressively improved solutions over iterations. The optimization alternates between the sampling and policy update phases across multiple epochs until the maximum number of epochs is reached. This iterative process enables DDRL to generate high-quality TSP solutions by combining the strengths of diffusion models and reinforcement learning.

In this framework, the latent vector $\phi$ contributes to generating TSP solution images via the diffusion denoising process described in Equation Equation 2. Simultaneously, $\phi$ defines an adjacency matrix that determines the final tour as a sequence of city indices. Throughout the diffusion denoising trajectory, the policy is updated to minimize the total tour length. Figure 3 illustrates this process, highlighting the interaction between diffusion-based denoising and RL-based policy optimization, which leads to increasingly optimized solutions. The complete procedure is described in Algorithm 1.

## 4 EXPERIMENTS

We conducted experiments across various city sizes, specifically $N = 20, 50, 100$, and $200$, to validate the generalization capability of our model. Using the Concorde solver as an Oracle for comparison, we evaluated DDRL against several baselines: a heuristic 2-opt algorithm, a transformer-based RL model (Kool et al., 2018; Bresson & Laurent, 2021), a GNN model trained with supervised learning (Joshi et al., 2019), and a diffusion-based method, Diffusion 50 (Graikos et al., 2022). We report results using objective values (Obj) and percentage gaps (Gap%), with Oracle exhibiting zero gaps as a benchmark.

### 4.1 BASIC TSP

As shown in Table 1, DDRL consistently achieves superior performance across all problem sizes in the Basic TSP setting. For $N = 20$, DDRL obtains an objective value of 3.84 with a minimal gap of 0.10%, effectively matching Oracle's solution. This high performance is sustained as the

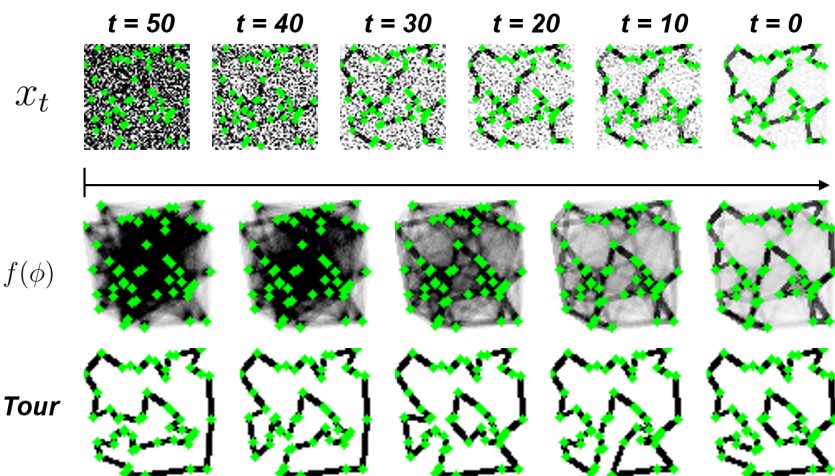

Figure 3: The visualization of (Top) the image-level view where diffusion denoising constructs the RL trajectory, and (Bottom) the graph-level perspective where RL optimization refines the latent vector $\phi$. This optimized $\phi$ generates an adjacency matrix, allowing optimization of both $f(\phi)$ and the graph tour.

---

**Algorithm 1** DDRL Optimization

---

1: **Input:** Latent vector $\phi$, prior knowledge $\epsilon_{\text{prior}}$, city positions $P \in \mathbb{R}^2$, constraints $c$
2: **Output:** Optimized tour solution $T^*$
3: $x_T \sim \mathcal{N}(0, \mathbf{I})$
4: **for** epoch $E = 1$ to $E_{\max}$ **do**
5:     **Sampling Phase:**
6:     **for** sample $i = 1$ to $N_{\text{samples}}$ **do**
7:         $\phi \leftarrow \arg\min_\phi \|\epsilon - \epsilon_{\text{prior}}(x_t, \phi)\|^2$           ▷ Initialization (Equation 5)
8:         $\gamma_i = \{x_T, x_{T-1}, \ldots, x_0\}$         ▷ Compute trajectory (Equation 2)
9:         Apply 2-opt local search to further refine $T_i$
10:        $r(\phi_i) = \text{reward\_function}(T_i, P, c)$       ▷ Calculate reward (Equation 3)
11:     **end for**
12:     Compute advantage $A_i = r_i - \bar{r}$, where $\bar{r}$ is the mean reward
13:     **Policy Update Phase:**
14:     **for** inner epoch $E_{\text{inner}} = 1$ to $E_{\text{inner\_max}}$ **do**
15:         **for** sample $i = 1$ to $N_{\text{samples}}$ **do**
16:            $\phi \leftarrow \phi + \alpha \nabla_\phi J(\phi)$         ▷ Update policy parameters (Equation 6)
17:         **end for**
18:     **end for**
19: **end for**

---

problem size increases, with objective values of 5.70, 7.83, and 10.94 for $N = 50$, $N = 100$, and $N = 200$, respectively. The results illustrate DDRL's robust generalization capability and scalability, maintaining gaps below 2.02% across all problem sizes. In contrast, supervised learning methods like Diffusion 50 (Graikos et al., 2022) struggle with larger problem sizes, highlighting DDRL's adaptability to varying environments.

## 4.2 CONSTRAINT SETTING

We generated TSP datasets for evaluation and created additional constraint-based datasets inspired by VanDrunen et al. (2023). These constraint datasets were designed to be visually intuitive yet challenging for conventional algorithms. As shown in Figure 4, we categorized the tasks into four types: basic TSP, Obstacle constraints, Path constraints, and Cluster constraints. In the **Basic TSP**, the goal is to find the shortest distance between cities without any constraints. For the **Obstacle Constraint**, penalties are imposed when paths are created within a rectangular area that violates

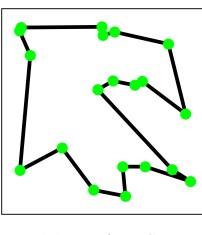 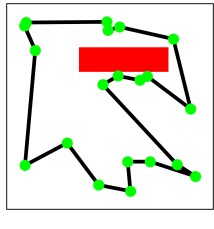 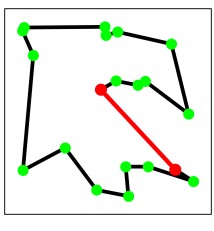 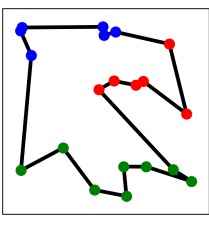

| (a) Basic TSP | (b) Obstacle Const. | (c) Path Const. | (d) Cluster Const. |

Figure 4: Examples of basic and constraint datasets. The green points represent city locations, and the black lines indicate connections between cities. (a) Basic TSP: No constraints; the goal is to find the shortest tour that visits all cities. (b) Obstacle Constraint: The red box represents a restricted area where no connections can be formed. (c) Path Constraint: The red lines represent predetermined paths that must be part of the tour. (d) Cluster Constraint: Cities are grouped by color, with the constraint that connections between clusters can occur only once, and round trips between clusters are not allowed.

| Algorithms | $N = 20$ | | $N = 50$ | | $N = 100$ | | $N = 200$ | |
|---|---|---|---|---|---|---|---|---|
| | Obj | Gap(%) | Obj | Gap(%) | Obj | Gap(%) | Obj | Gap(%) |
| Oracle | **3.84** | **0** | **5.69** | **0** | **7.759** | **0** | **10.72** | **0** |
| 2-opt | 3.93 | 2.24 | 5.86 | 2.95 | 8.03 | 3.54 | 11.69 | 9.07 |
| Transformer (Kool et al., 2018) | 3.85 | 0.24 | 5.80 | 1.76 | 8.12 | 4.53 | 11.24 | 7.18 |
| GNN (Joshi et al., 2019) | 3.86 | 0.60 | 5.87 | 3.10 | 8.41 | 8.38 | 13.45 | 25.52 |
| Transformer (Bresson & Laurent, 2021) | 3.85 | 0.29 | 5.71 | 0.31 | 7.88 | 1.42 | 12.38 | 15.50 |
| Diffusion 50 (Graikos et al., 2022) | 3.89 | 1.16 | 5.76 | 1.28 | 7.92 | 2.13 | 11.21 | 4.64 |
| DDRL | **3.84** | **0.10** | **5.70** | **0.13** | **7.83** | **0.87** | **10.94** | **2.02** |

Table 1: Performance Comparison of TSP Algorithms Across Different Problem Sizes (Basic). DDRL consistently achieves the lowest gaps across all problem sizes, indicating its strong generalization capability. Values are rounded to two decimal places. For $N = 20$, DDRL's objective value matches Oracle's, but a gap is still present due to slight differences in the values beyond the second decimal place.

specified conditions. The **Path Constraint** requires traversal through predetermined paths with a range of 1 to 4, meaning no other path can pass through them. Lastly, the **Cluster Constraint** prioritizes connections within clusters before allowing inter-cluster connections, with the restriction that inter-cluster links can only occur once.

For the baseline models, additional rules were applied as they could not infer constraint conditions under the default settings. In the transformer-based models (Kool et al., 2018; Bresson & Laurent, 2021), the autoregressive decoding process was adjusted to ensure that constraints were satisfied. In the GNN-based model (Joshi et al., 2019), we modified the beam search by setting the path connection probabilities to 1 or 0, depending on the constraint, to ensure compliance. Similarly, for the diffusion-based model (Graikos et al., 2022), we added rules during the 2-opt optimization process, as was done with DDRL, to meet the constraint conditions. Under the constraint conditions, the cost is calculated as the sum of the total path length and the penalty cost. The penalty cost is computed as the product of the penalty constant and the penalty violation count, with the penalty constant set to one in this experiment.

The experimental results indicate that DDRL outperforms existing approaches in both basic TSP problems and under the diverse constraint conditions described. It shows that DDRL is versatile and robust, capable of effectively addressing TSP problems with complex constraints.

### 4.3 OBSTACLE CONSTRAINT

Table 2 demonstrates DDRL's ability to handle TSP instances with obstacle constraints effectively.

| Algorithms | N = 20 | | N = 50 | | N = 100 | | N = 200 | |
|---|---|---|---|---|---|---|---|---|
| | Obj | Gap(%) | Obj | Gap(%) | Obj | Gap(%) | Obj | Gap(%) |
| Oracle | **4.16** | **0** | **5.89** | **0** | **7.87** | **0** | **10.77** | **0** |
| 2-opt | 14.56 | 250.55 | 32.93 | 460.28 | 62.06 | 689.16 | 118.83 | 1004.00 |
| Transformer (Kool et al., 2018) | 6.39 | 53.51 | 9.97 | 69.42 | 12.83 | 63.01 | 20.09 | 86.61 |
| GNN (Joshi et al., 2019) | 4.72 | 13.24 | 7.01 | 19.04 | 9.48 | 20.43 | 20.85 | 93.35 |
| Transformer (Bresson & Laurent, 2021) | 6.74 | 62.05 | 7.82 | 32.86 | 10.53 | 33.77 | 26.25 | 143.75 |
| Diffusion 50 (Graikos et al., 2022) | 5.09 | 22.21 | 7.02 | 19.23 | 8.91 | 13.23 | 12.30 | 14.18 |
| DDRL | **4.21** | **1.07** | **5.94** | **0.96** | **8.08** | **2.65** | **11.15** | **3.53** |

Table 2: Performance comparison of TSP algorithms with Obstacle Constraints across different problem sizes. DDRL consistently outperforms other models, demonstrating superior handling of geometric constraints and effectively avoiding overlaps. This highlights the strength of DDRL in solving visually complex TSP instances.

| Algorithms | N = 20 | | N = 50 | | N = 100 | | N = 200 | |
|---|---|---|---|---|---|---|---|---|
| | Obj | Gap(%) | Obj | Gap(%) | Obj | Gap(%) | Obj | Gap(%) |
| Oracle | **4.32** | **0** | **6.18** | **0** | **8.01** | **0** | **10.72** | **0** |
| 2-opt | 5.85 | 34.15 | 7.09 | 14.47 | 15.69 | 95.81 | 107.31 | 881.81 |
| Transformer (Kool et al., 2018) | 4.62 | 7.00 | 7.46 | 20.49 | 11.03 | 37.61 | 21.07 | 92.75 |
| GNN (Joshi et al., 2019) | 6.57 | 51.34 | 8.63 | 39.10 | 10.02 | 24.33 | 19.55 | 80.34 |
| Transformer (Bresson & Laurent, 2021) | 4.86 | 12.05 | 6.91 | 11.42 | 9.00 | 11.04 | 18.03 | 67.93 |
| Diffusion 50 (Graikos et al., 2022) | 5.47 | 25.84 | 7.22 | 16.45 | 8.67 | 8.20 | 11.91 | 9.19 |
| DDRL | **4.57** | **5.33** | **6.40** | **3.50** | **8.12** | **1.31** | **11.17** | **2.59** |

Table 3: Performance comparison of TSP algorithms with Path Constraint across various problem sizes. DDRL excels at generating accurate solutions while satisfying path constraints and achieving lower gaps compared to baseline models. These results emphasize DDRL's effectiveness in handling predefined route conditions.

DDRL consistently outperforms baseline methods, achieving a low objective value of 5.94 and a gap of 0.96% for $N = 50$. Unlike 2-opt and Transformer-based models (Kool et al., 2018), (Bresson & Laurent, 2021), which suffer from significant performance degradation as the problem size grows, DDRL maintains strong scalability and accuracy. Even when compared to diffusion-based methods like Diffusion 50 (Graikos et al., 2022), DDRL delivers better overall results, showing that it leverages visual constraints effectively while retaining high performance across all problem sizes.

### 4.4 PATH CONSTRAINT

In Table 3, DDRL demonstrates its versatility by achieving competitive performance under path constraints, especially in larger-scale problems. For $N = 50$, DDRL produces an objective value of 6.40 with a gap of only 3.50%, outperforming both transformer-based (Kool et al., 2018), (Bresson & Laurent, 2021) and GNN models (Joshi et al., 2019) as the problem size increases. DDRL's generalization capability is evident as it maintains low gaps even with larger city sizes ($N = 100$ and $N = 200$), while other models exhibit substantial declines in performance. DDRL's ability to navigate predefined path conditions further emphasizes its adaptability and efficiency.

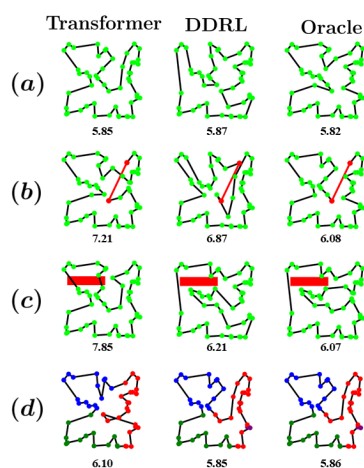

Figure 5: Comparison of inference processes. Each row is a sample from a basic TSP, Obstacle Constraint, and Path Constraint, in that order. The below numbers are Obj with a penalty of 1.

| Algorithms | N = 20 | | N = 50 | | N = 100 | | N = 200 | |
|---|---|---|---|---|---|---|---|---|
| | **Obj** | **Gap(%)** | **Obj** | **Gap(%)** | **Obj** | **Gap(%)** | **Obj** | **Gap(%)** |
| Oracle | **3.91** | **0** | **5.82** | **0** | **7.97** | **0** | **11.03** | **0** |
| 2-opt | 4.67 | 19.33 | 8.42 | 44.60 | 12.09 | 51.89 | 15.97 | 45.35 |
| Transformer (Kool et al., 2018) | 4.96 | 26.69 | 10.18 | 74.75 | 20.72 | 160.22 | 27.20 | 147.51 |
| GNN (Joshi et al., 2019) | 4.56 | 16.59 | 8.68 | 49.09 | 16.70 | 109.72 | 25.64 | 133.25 |
| Transformer (Bresson & Laurent, 2021) | 4.16 | 6.48 | 7.19 | 23.43 | 12.68 | 59.22 | 24.07 | 119.00 |
| Diffusion 50 (Graikos et al., 2022) | 4.57 | 16.57 | 7.89 | 35.29 | 11.49 | 44.26 | 15.69 | 43.00 |
| DDRL | **4.05** | **3.54** | **6.51** | **11.69** | **10.52** | **32.05** | **15.56** | **41.55** |

Table 4: Performance comparison of TSP algorithms with Cluster Constraint across different problem sizes. DDRL achieves the best performance, especially for larger instances, demonstrating its scalability and robustness in handling clustered constraints, where baseline models struggle to maintain effectiveness as city counts increase.

## 4.5 CLUSTER CONSTRAINT

As seen in Table 4, DDRL excels in TSP problems with cluster constraints, consistently outperforming other models, particularly as problem complexity increases. For $N = 200$, DDRL achieves the best performance with an objective value of 15.44 and a gap of 40.38%, demonstrating superior scalability compared to GNN (Joshi et al., 2019) and transformer models (Kool et al., 2018), (Bresson & Laurent, 2021), which struggle under these conditions. While diffusion-based models like Diffusion 50 (Graikos et al., 2022)show competitive performance, DDRL continues to outperform them in handling inter-cluster complexity, showcasing its robustness and ability to handle clustered constraints effectively in larger-scale problems.

## 4.6 QUALITATIVE ANALYSIS

Figure 5 visualizes the inferred solution of the baseline method (Bresson & Laurent, 2021), DDRL, and Oracle. Each row denotes the baseline TSP, Obstacle Constraint, and Path Constraint, respectively. DDRL shows superior performance on all tasks without obstacles and path constraints. We performed an ablation study to analyze the impact of prior knowledge and initialization techniques on DDRL's performance. Figure 6 demonstrates that when both elements are applied, the model converges significantly faster than when either one or both are omitted. The pre-trained diffusion model, serving as prior knowledge, effectively guides the connections between cities, while the initialization technique ensures a well-formed adjacency matrix from the start. These combined factors enhance the early-stage learning stability, enabling faster convergence and improved overall performance.

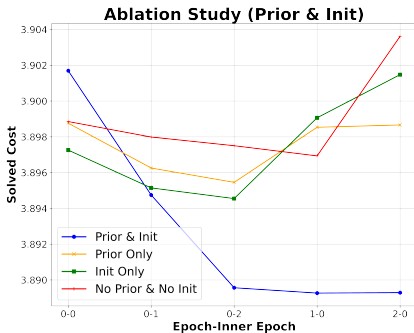

Figure 6: Ablation study showing the impact of prior knowledge and initialization on the convergence speed in DDRL training.

## 5 CONCLUSION

This paper introduced DDRL, a novel approach integrating diffusion models with RL to address the TSP. By leveraging the strengths of both graph and image representations, DDRL effectively handles both basic and complex constraint-based TSP instances. Our method demonstrates superior scalability, generalization, and convergence stability compared to state-of-the-art algorithms, benefiting from the incorporation of pre-trained diffusion models as prior knowledge. Extensive experiments show that DDRL achieves state-of-the-art performance, making it a promising solution for large-scale combinatorial optimization problems. Additionally, DDRL proves effective across various TSP variants with different constraints, demonstrating its adaptability and robustness in handling complex and diverse optimization tasks.

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

# Supplementary Materials for
# DDRL: A Diffusion-Driven Reinforcement Learning Approach for Enhanced TSP Solutions

## A    ADDITIONAL QUALITATIVE ANALYSIS

In this section, we present detailed visualizations of the TSP solutions generated under various conditions: Basic, Obstacle constraint, Path constraint and Cluster constraint. These visualizations are provided for different city sizes ($N = 20, 50, 100$). Figures 7 to 9 showcase the visual representations of TSP instances produced by the DDRL model.

Each figure is structured into four main columns corresponding to the constraint types:

- **Basic**: Visualizes the basic TSP solution without any additional constraints.
- **Obstacle**: Illustrates the TSP solution under an obstacle constraint, requiring the solution to navigate around specific blocked areas or paths.
- **Path**: Displays the TSP solution under a path constraint, where the tour must pass through specific points or follow a particular route.
- **Cluster**: Demonstrates the TSP solution under a cluster constraint, where cities are grouped into clusters, and the solution must visit each cluster exactly once.

For each constraint type, the visualizations are further divided into four key elements:

1. **Ground Truth**: The leftmost image represents the ground truth of the city distribution and the optimal TSP tour.
2. **Latent($\mathbf{x}_t$)**: The second image from the left shows the image $x_t$, obtained during the diffusion denoising process.
3. **Encoding($\mathbf{f}(\phi)$)**: The third image from the left visualizes the encoding of the features $f(\phi)$, capturing the problem's structural information.
4. **Solved Tour**: The rightmost image represents the TSP solution output by the model under the given constraints.

These visualizations collectively demonstrate the DDRL model's capability to generalize across different problem sizes and conditions, not only solving the basic TSP but also adapting to more complex scenarios with obstacles, path and cluster constraint. This generalization is crucial for real-world applications where additional constraints often complicate routing problems.

Moreover, the figures illustrate that the model can effectively infer solutions under various constraint conditions, maintaining near-optimal performance even in the presence of significant obstacles or mandatory paths or clustered cities. This robustness highlights the model's potential for broader applicability beyond traditional TSP scenarios.

## B    HYPERPARAMETER DESCRIPTION

In this section, we detail the key hyperparameters used in the DDRL model, focusing on their derivation, role, and usage within the model.

### B.1    NOISE SCHEDULE PARAMETER ($\alpha_t$)

The $\alpha_t$ used in DDRL follows the definition from DDIM Song et al. (2020). In DDIM, $\alpha_t$ is defined as the product of $1 - \beta$:

$$\alpha_t = \prod_{s=1}^{t} (1 - \beta_s)$$

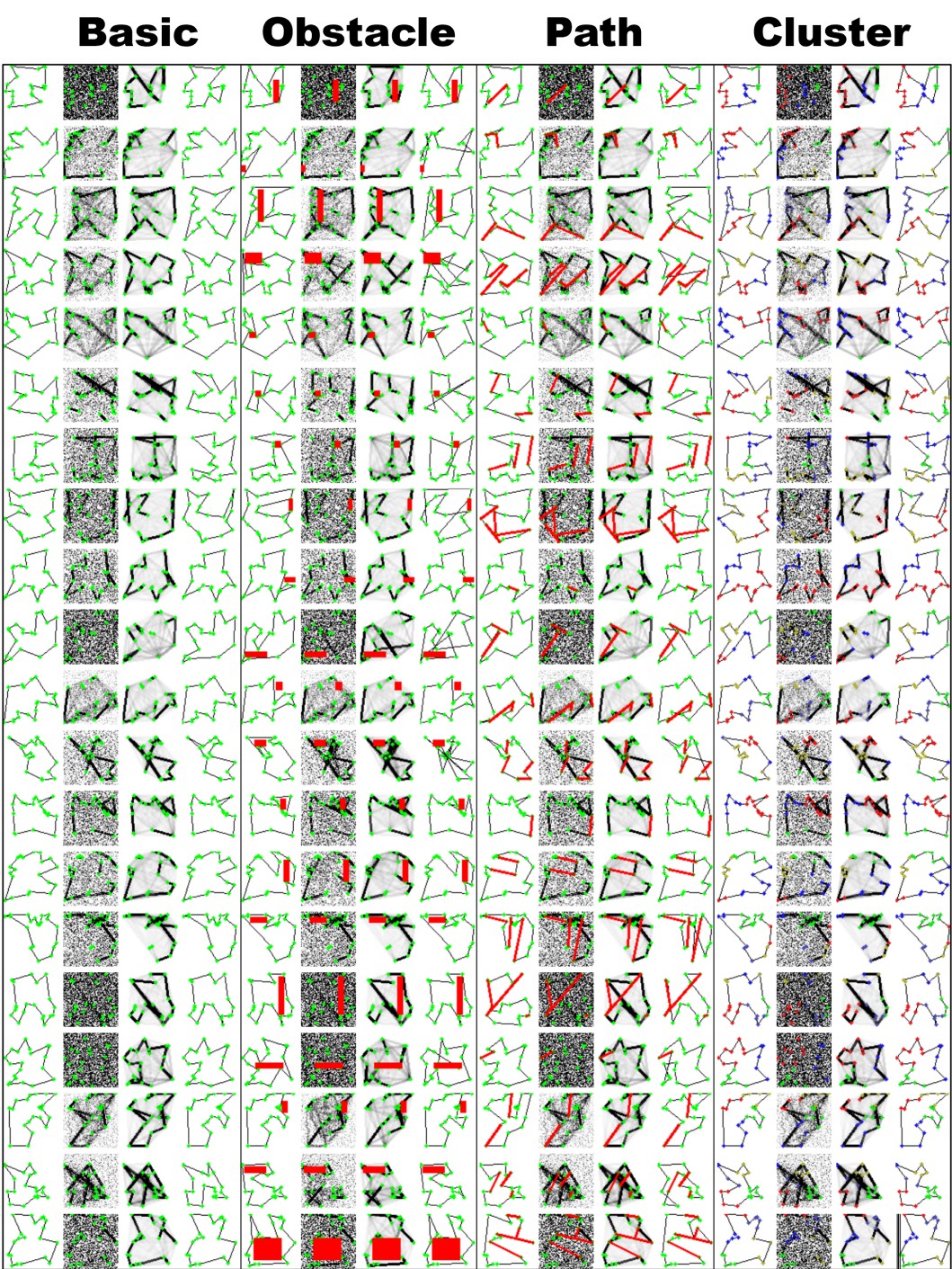

Figure 7: Visualization of outputs for Basic, Obstacle, Path and Cluster scenarios when $N = 20$.

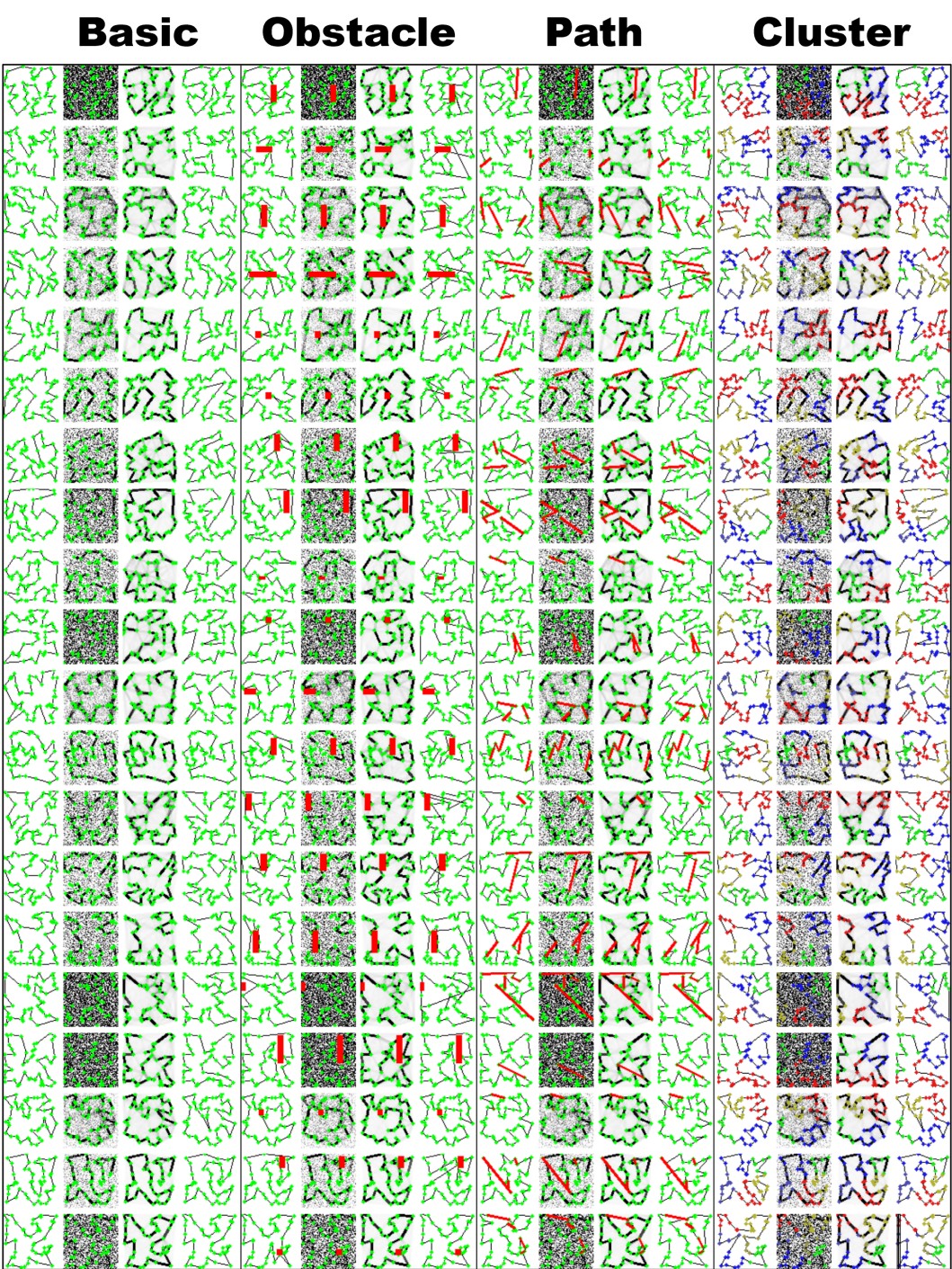

Figure 8: Visualization of outputs for Basic, Obstacle, Path and Cluster scenarios when $N = 50$.

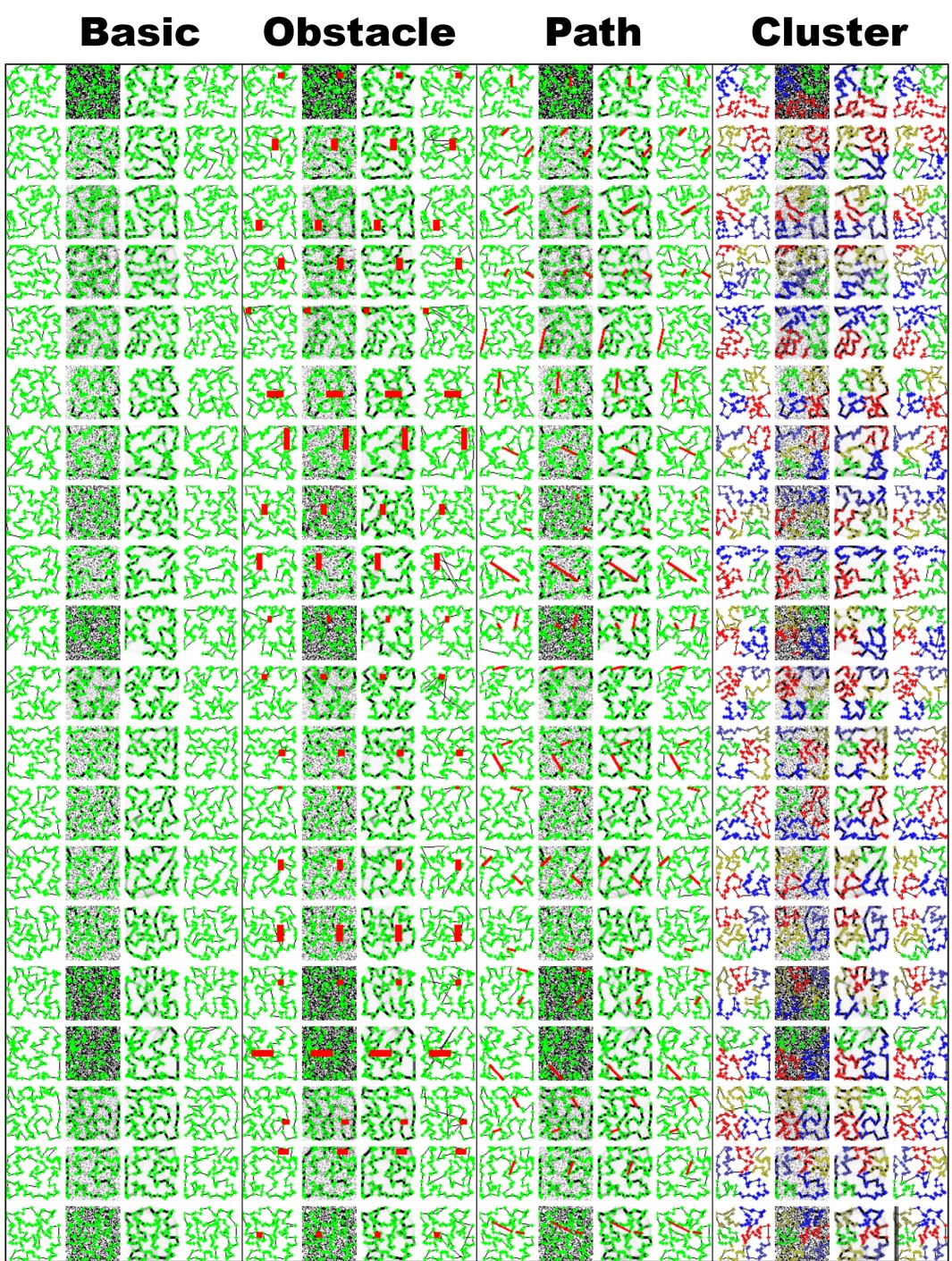

Figure 9: Visualization of outputs for Basic, Obstacle, Path and Cluster scenarios when $N = 100$.

As in DDIM, $\alpha_t$ in DDRL plays a crucial role in balancing noise introduction during the forward process and its removal during the reverse process. By adjusting $\alpha_t$, DDRL effectively transitions from noisy data to high-quality images, enabling efficient sampling and ensuring robust performance across various scenarios.

## B.2 Noise Variance ($\sigma_t^2$)

The noise variance $\sigma_t^2$ in the diffusion process is directly related to the $\alpha_t$ parameter defined earlier. It quantifies the uncertainty during the denoising process and is crucial for managing the trade-off between exploration and exploitation. Specifically, $\sigma_t^2$ is derived from the relationship between the $\alpha_t$ values at consecutive timesteps:

$$\sigma_t^2 = \eta^2 \cdot \left( \frac{(1 - \alpha_{t-1})}{(1 - \alpha_t)} \right) \cdot \left( 1 - \frac{\alpha_t}{\alpha_{t-1}} \right)$$

Here, $\alpha_t$ represents the cumulative product of $(1 - \beta_s)$ up to timestep $t$, as defined earlier. The noise variance $\sigma_t^2$ plays a crucial role in the denoising process by allowing the model to introduce controlled randomness during each diffusion step. This helps the model explore diverse solutions while progressively refining the quality of the generated samples, ensuring effective and robust performance across different scenarios.

## B.3 Modified Cumulative Noise Schedule ($\overline{\alpha_{t_i}}$)

In DDRL, the modified cumulative noise schedule $\overline{\alpha_{t_i}}$ is derived from the noise variance $\beta_t$ and is calculated as the cumulative product of $1 - \beta_s$ across all timesteps $s$ up to $t_i$:

$$\overline{\alpha_{t_i}} = \prod_{s=1}^{t_i} (1 - \beta_s)$$

While this formula is mathematically similar to the $\alpha_t$ defined in subsection B.1, the role of $\overline{\alpha_{t_i}}$ in DDRL is distinct. Specifically, $\overline{\alpha_{t_i}}$ is used during the initialization phase to stabilize the latent vector $\phi$ by leveraging a pretrained diffusion model (prior knowledge) and diffusion loss. This contrasts with the use of $\alpha_t$ in the RL-based denoising process, where $\alpha_t$ manages the noise introduction and removal during sampling. By distinguishing $\overline{\alpha_{t_i}}$ from $\alpha_t$, DDRL effectively handles the different demands of initialization and sampling, ensuring robust and efficient performance across various scenarios.

## C  Dataset Generation Process

The goal of generating these datasets is to evaluate the performance of TSP-solving algorithms in the presence of additional constraints, beyond the basic TSP problem. By incorporating constraint conditions, we can assess how well these algorithms adapt and maintain their efficiency under more complex scenarios.

Three types of constraint conditions were introduced, and for each of them, the constraints were applied to the same city locations as those in the basic dataset. These constraints inevitably increase the overall path length compared to the default setting, providing a basis for comparison between the results from basic and constrained TSP instances.

The dataset generation process was formulated by adding penalties to the city-to-city distances in the basic setting. By assigning sufficiently large penalties, the priority between paths is adjusted, allowing the solver to find solutions that satisfy the constraint conditions. However, adding constraints increases the complexity of the TSP problem, which introduces scaling limitations when using solvers. As the complexity increases with the number of cities and the number of constraints, the cost of generating valid data also rises significantly. Of course, DDRL and other approximation algorithms offer the advantage of relatively faster inference, even in large-scale settings.

## C.1 BASIC DATASET

The basic dataset consists of 1,280 instances for each city size. This dataset is sourced from the repository available at `https://github.com/chaitjo/learning-tsp`. Additionally, the prior knowledge used in our approach, specifically the pretrained diffusion model, was obtained from `https://diffusion-priors.s3.amazonaws.com/unet50_64_8.pth`.

## C.2 OBSTACLE CONSTRAINT DATASET GENERATION FOR TSP

This section outlines the method for generating a dataset with obstacle constraints for the TSP. The process begins with an existing dataset, $D_{\text{basic}}$, which contains $N$ cities with fixed coordinates $P \in \mathbb{R}^{N \times 2}$ and corresponding ground truth tours $T_{\text{GT}}$. Our goal is to extend this dataset to produce $D_{\text{obstacle}}$, which includes the original city information, obstacle information $B$ (defined as a box), and optimal tours $R$ that respect these obstacle constraints.

For each city in $D_{\text{basic}}$, we attempt to find an optimal obstacle box $B_{\text{opt}}$ that maximizes overlap with the ground truth tour $T_{\text{GT}}$, while ensuring no city points are inside the box. Maximizing this overlap increases the obstacle's influence on the tour, allowing us to assess how well models can adapt to the changes introduced by the obstacle. Once an optimal box is found, we compute the distance matrix $\mathbf{M}$, which penalizes the distances between city pairs whose connecting paths intersect the obstacle. Solving the TSP with this modified matrix prevents the solution from passing through the obstacle, ensuring that the generated dataset respects the obstacle constraints. In this paper, we set the penalty value to 100.

If the resulting tour $S$ respects the obstacle constraints, it is saved along with the obstacle information. If no valid solution is found, random obstacle boxes $B_{\text{rand}}$ are generated until a valid tour is obtained. This ensures that each city in the dataset has an associated tour that adheres to the imposed constraints. The new dataset, $D_{\text{obstacle}}$, thus includes the city coordinates, obstacle box coordinates (top-left and bottom-right), and the optimal tour that respects the constraints. The overall process for generating the dataset is outlined in Algorithm 2.

---

**Algorithm 2** Obstacle Constraints Dataset Generation for TSP

---

1: **Input:** $D_{\text{basic}}$ (Dataset of $N$ cities containing city coordinates $P \in \mathbb{R}^{N \times 2}$ and ground truth tours $T_{\text{GT}}$)
2: **Output:** $D_{\text{obstacle}}$ (Dataset with added obstacle information $B$ and optimal tour information $R$)
3: **for** each $i \in \{1, 2, \ldots, N\}$ **do**
4:     $P_i \leftarrow$ City coordinates from $D_{\text{basic}}$ ($P_i \in \mathbb{R}^2$)
5:     $T_{\text{GT}_i} \leftarrow$ Ground truth tour from $D_{\text{basic}}$
6:     $R_i \leftarrow \emptyset, B_i \leftarrow \emptyset$
7:     $B_{\text{opt}} \leftarrow$ find_optimal_box($P_i, T_{\text{GT}_i}$)
8:     **if** $B_{\text{opt}} \neq \emptyset$ **then**
9:         $\mathbf{M} \leftarrow$ calculate_distance_matrix($P_i, B_{\text{opt}}$)
10:         $S \leftarrow$ solve_tsp($\mathbf{M}$)                                    ▷ using Concorde solver
11:         **if** is_valid($S, P_i, B_{\text{opt}}$) **then**
12:             $R_i \leftarrow S, B_i \leftarrow B_{\text{opt}}$
13:         **end if**
14:     **end if**
15:     **while** $R_i = \emptyset$ **do**
16:         $B_{\text{rand}} \leftarrow$ generate_random_box($T_{\text{GT}_i}$)
17:         $\mathbf{M} \leftarrow$ calculate_distance_matrix($P_i, B_{\text{rand}}$)
18:         $S \leftarrow$ solve_tsp($\mathbf{M}$)                                    ▷ using Concorde solver
19:         **if** is_valid($S, P_i, B_{\text{rand}}$) **then**
20:             $R_i \leftarrow S, B_i \leftarrow B_{\text{rand}}$
21:         **end if**
22:     **end while**
23:     Save $P_i, R_i, B_i$ to $D_{\text{obstacle}}$
24: **end for**

---

**Function Descriptions** `find_optimal_box`: This function finds the obstacle box that maximizes overlap with the ground truth tour $T_{\text{GT}}$. The purpose of maximizing the overlap is to increase the influence of the obstacle on the tour, thereby testing the model's ability to find an optimal solution in a modified setting. The function evaluates possible obstacle boxes based on their overlap with $T_{\text{GT}}$, selecting the box that maximizes overlap while avoiding city points.

`generate_random_box`: This function generates a random obstacle box around a segment of the ground truth tour, ensuring no city points are inside. It serves as a fallback mechanism when an optimal box is not found.

`calculate_distance_matrix`: This function computes the distance matrix $\mathbf{M}$ for city pairs, with adjustments for obstacle constraints. If a path between two cities intersects the obstacle box, a penalty of 100 is added to the distance between those cities. This encourages the solver to avoid obstacle-affected paths, ensuring that the generated solution respects the obstacle constraints.

`is_valid`: This function checks if the computed tour from the new TSP setting (with constraints) intersects the obstacle box. If any segment of the tour crosses the obstacle, the solution is invalid; otherwise, it is valid.

### C.3  PATH CONSTRAINT DATASET GENERATION FOR TSP

This section outlines the method for generating a dataset with path constraints for the TSP. Starting with the existing dataset $D_{\text{basic}}$, which contains $N$ cities with fixed coordinates $P \in \mathbb{R}^{N \times 2}$ and corresponding ground truth tours $T_{\text{GT}}$, we extend this dataset to produce $D_{\text{path}}$. The output dataset includes the original city information, sampled path constraint information $E$ (defined as predetermined paths between cities), and optimal tours $R$ that respect these predetermined paths.

For each city in $D_{\text{basic}}$, we sample a set of predetermined paths $E_{\text{sample}}$ from paths that are not part of the ground truth tour $T_{\text{GT}}$. These predetermined paths are selected to ensure that there are no intersections between them. Once valid predetermined paths are found, we compute the distance matrix $\mathbf{M}$, where penalties are imposed on all paths not included in the predetermined paths. This encourages the solution to prioritize the use of the predetermined paths when solving the TSP. Solving the TSP with this modified matrix produces a tour $S$ that respects the path constraints. If the resulting tour is valid, meaning it does not have any intersections and adheres to the predetermined paths, it is saved along with the corresponding path information $E_{\text{sample}}$. This ensures that each city in the dataset has an associated tour that adheres to the imposed path constraints.

The number of predetermined paths depends on the number of cities. As the number of cities increases, the number of predetermined paths also grows, which increases the complexity of dataset generation. When there are many cities, having too many predetermined paths further complicates the dataset generation task. Therefore, to manage this complexity, the maximum number of predetermined paths is reduced as the number of cities increases. The overall process for generating the dataset is outlined in Algorithm 3.

**Function Descriptions** `sampling_edge`: This function samples a set of paths that are not part of the ground truth tour $T_{\text{GT}}$. The paths are selected to ensure that there are no intersections between them. The purpose of sampling paths not in the ground truth tour is to evaluate how well models adapt to new constraints and deviate from the basic TSP setting.

`calculate_distance_matrix`: This function computes the distance matrix $\mathbf{M}$ for the city points. Penalties are added to the distances between cities if their connecting path is not part of the predetermined paths. The penalty ensures that the solver prioritizes the predetermined paths when computing the optimal tour. In this paper, the penalty value is set to 100.

`check_tour_intersections`: This function checks for intersections between the paths in a given tour. It is not limited to the tours computed by the TSP solver; it can also be applied to any arbitrary set of tours to check for intersections between paths.

### C.4  CLUSTER CONSTRAINT DATASET GENERATION FOR TSP

This section describes the process for generating a dataset with cluster constraints for the TSP. Starting with an existing dataset, $D_{\text{basic}}$, which contains $N$ cities with fixed coordinates $P \in \mathbb{R}^{N \times 2}$ and

---

**Algorithm 3** Path Constraint Dataset Generation for TSP

---

1: **Input:** $D_{\text{basic}}$ (Dataset of $N$ cities containing city coordinates $P \in \mathbb{R}^{N \times 2}$ and ground truth tours $T_{\text{GT}}$)
2: **Output:** $D_{\text{path}}$ (Dataset with path constraint information and optimal tours)
3: **for** each $i \in \{1, 2, \dots, N\}$ **do**
4:     $P_i \leftarrow$ City coordinates from $D_{\text{basic}}$ ($P_i \in \mathbb{R}^2$)
5:     $T_{\text{GT}_i} \leftarrow$ Ground truth tour from $D_{\text{basic}}$
6:     $S \leftarrow \emptyset, E_i \leftarrow \emptyset$
7:     **while** $S = \emptyset$ **do**
8:         $E_{\text{sample}} \leftarrow \text{sampling\_edge}(T_{\text{GT}_i}, P_i)$
9:         $\mathbf{M} \leftarrow \text{calculate\_distance\_matrix}(P_i, E_{\text{sample}})$
10:        $S \leftarrow \text{solve\_tsp}(\mathbf{M})$                            ▷ using Concorde solver
11:        **if** $\text{check\_tour\_intersections}(S, P_i)$ **then**
12:           $S \leftarrow \emptyset$
13:        **else**
14:           $E_i \leftarrow E_{\text{sample}}$
15:        **end if**
16:     **end while**
17:     Save $P_i$, $S$, $E_i$ to $D_{\text{path}}$
18: **end for**

---

corresponding ground truth tours $T_{\text{GT}}$, we extend this dataset to produce $D_{\text{cluster}}$. The output dataset includes the original city information, cluster information $C$ (clusters assigned by a clustering algorithm like KMeans), and optimal tours $R$ that respect the cluster constraints.

For each instance in $D_{\text{basic}}$, we apply a clustering algorithm (such as KMeans) to assign cities to $k$ clusters. The value of $k$ is dynamically chosen based on the properties of each instance. Once the clusters are assigned, we adjust the distance matrix $\mathbf{M}$ by adding a penalty to the distances between cities in different clusters. This penalty encourages the solver to prioritize connections within the same cluster. Solving the TSP with the adjusted distance matrix generates a tour $S$ that aims to respect the cluster constraints. The solution is validated by ensuring that it adheres to the cluster constraints, particularly maintaining the correct in-degree and out-degree for the clusters. Once a valid tour is obtained, the dataset is saved with the cluster assignments and corresponding tour information. The overall process is outlined in Algorithm 4.

---

**Algorithm 4** Cluster Constraint Dataset Generation for TSP

---

1: **Input:** $D_{\text{basic}}$ (Dataset of $N$ cities containing city coordinates $P \in \mathbb{R}^{N \times 2}$ and ground truth tours $T_{\text{GT}}$)
2: **Output:** $D_{\text{cluster}}$ (Dataset with cluster information $C$ and optimal tours)
3: **for** each $i \in \{1, 2, \dots, N\}$ **do**
4:     $P_i \leftarrow$ City coordinates from $D_{\text{basic}}$ ($P_i \in \mathbb{R}^2$)
5:     $T_{\text{GT}_i} \leftarrow$ Ground truth tour from $D_{\text{basic}}$
6:     $S \leftarrow \emptyset, C_i \leftarrow \emptyset$
7:     **while** $S = \emptyset$ **do**
8:         $k \leftarrow \text{select\_cluster\_number}(P_i)$
9:         $C_{\text{sample}} \leftarrow \text{perform\_clustering}(P_i, k)$                   ▷ using KMeans
10:        $\mathbf{M} \leftarrow \text{calculate\_distance\_matrix}(P_i, C_{\text{sample}})$
11:        $S \leftarrow \text{solve\_tsp}(\mathbf{M})$                           ▷ using Concorde solver
12:        **if** $\text{check\_cluster\_violations}(S, C_{\text{sample}})$ **then**
13:           $S \leftarrow \emptyset$
14:        **else**
15:           $C_i \leftarrow C_{\text{sample}}$
16:        **end if**
17:     **end while**
18:     Save $P_i$, $S$, $C_i$ to $D_{\text{cluster}}$
19: **end for**

---

**Function Descriptions**  `calculate_distance_matrix`: This function modifies the distance matrix by adding a penalty to the distances between cities that belong to different clusters. This encourages the solver to prioritize connections within the same cluster when computing the optimal tour. In this paper, a penalty of 100 is applied for inter-cluster connections.

`check_cluster_violations`: This function verifies that the computed tour adheres to the cluster constraint by checking for violations in the in-degree and out-degree conditions of the clusters. The tour is considered valid if each cluster has exactly one in-degree and one out-degree connection, ensuring that no cluster is skipped or revisited unnecessarily.

## D  DIFFUSION POLICY GRADIENT

The gradient of the loss function is equivalent to the diffusion policy gradient as follows. This proposition is based on the assumption that $p_\phi(x_{0:T})r(\phi)$ and its derivative with respect to $\phi$ are continuous relative to $\phi$ and $x_{0:T}$, which permits the interchange between differentiation and integration.

$$
\begin{aligned}
\nabla_\phi J(\phi) &= \nabla_\phi \mathbb{E}_{x_t}[r(\phi)] \\
&= \nabla_\phi \int_{x_t} p_\phi(x_t)r(\phi)dx_t \\
&= \nabla_\phi \int \cdots \int p_\phi(x_{0:T})r(\phi)dx_{0:T} \\
&= \int \cdots \int p_\phi(x_{0:T})r(\phi)\nabla_\phi \log p_\phi(x_{0:T})dx_{0:T} \\
&= \mathbb{E}_{x_{0:T}}[r(\phi) \sum_{t=0}^{T-1} \nabla_\phi \log p_\phi(x_t|x_{t+1})] \\
&= \mathbb{E}_{x_{0:T}}[r(\phi) \sum_{t=1}^{T} \nabla_\phi \log p_\phi(x_{t-1}|x_t)] \\
&= \mathbb{E}_{s_{0:T}}[r(\phi) \sum_{\tau=0}^{T-1} \nabla_\phi \log \pi(a_\tau|s_\tau)].
\end{aligned}
$$

## E  EXPERIMENT DETAILS

This section outlines the key experimental setup used for training and evaluating the DDRL model.

### E.1  HARDWARE SETUP

Experiments were conducted on a system with the following specifications:

- **CPU**: Intel Core i9-10900X, 10 cores, 20 threads
- **GPU**: 4 x NVIDIA GeForce, RTX 4090 24 GiB VRAM each
- **Memory**: 188 GiB RAM
- **OS**: Ubuntu 20.04.6 LTS
- **Libraries**: PyTorch 2.1.2, CUDA 12.0

### E.2  REPRODUCIBILITY

To ensure consistency across all experiments, a fixed random seed was used:

- **Random Seed**: 2024

### E.3 TRAINING CONFIGURATION

The model was trained under the following conditions:

- **Training Epochs**: 3, 20, 30, 50
- **Inner Epochs per Training Epoch**: 3, 10
- **Initial Sample Size**: 3, 5, 10

These configurations were selected to optimize the balance between training efficiency and model performance, ensuring robust convergence without overfitting. For instances with relatively lower complexity, such as $N = 20$ or $N = 50$, longer training epochs (30 or more) were used to fully explore the solution space. In contrast, for more complex instances with $N = 100$ or higher, shorter training epochs (20 or fewer) were employed to maintain training efficiency while still achieving satisfactory performance.

