# OpenReview forum: "DDRL: A DIFFUSION-DRIVEN REINFORCEMENT LEARNING APPROACH FOR ENHANCED TSP SOLUTIONS"
_ICLR.cc/2025/Conference — ICLR 2025 Conference Withdrawn Submission_

### Official Review · Reviewer_w226 · 2024-10-20

**Soundness:** 2
**Presentation:** 3
**Contribution:** 2
**Rating:** 3
**Confidence:** 4

**Summary:**

This paper presents a method integrating diffusion models with reinforcement learning (RL) to solve Traveling Salesman Problems (TSPs). They use image to represent TSP, then optimize the latent matrix corresponding to the TSP image using RL.

**Strengths:**

1. The attempt of combining TSP images and RL seems interesting and original.
2. The structure of this paper is reasonable.

**Weaknesses:**

1. The proposed method may have limited applicability to other combinatorial optimization problems due to the following reasons:
   a) Images might only be effective in representing certain graph-based problems, which restricts their contribution to the broader field of neural combinatorial optimization.
   b) The use of images could significantly slow down inference speed.
   c) Image resolution may heavily impact model performance, further limiting the method's generalization.

2. The baseline models used in the study are outdated, with the most recent one published in 2022. Additionally, these baselines do not account for constraint modeling, potentially making the comparison experiments unfair.

3. The paper lacks a thorough discussion on computational cost, including inference and training time, as well as scalability when addressing larger TSP instances.

4. It is noted that the dimension of the latent matrix is predefined and fixed. This raises concerns about how the method generalizes to TSP instances of varying sizes. I recommend the authors conduct additional experiments using the TSPlib benchmarking dataset to address this issue.

**Questions:**

Please refer to the weakness section.

---

> ### Author Response · Authors · 2024-12-02
>
> We sincerely thank you for your detailed review and thoughtful feedback. We appreciate the opportunity to address your concerns regarding the applicability, experimental setup, computational costs, and generalization of DDRL.
>
> 1. Applicability of DDRL to Other Combinatorial Optimization Problems
>
>     a) Effectiveness of Images in Representing Graph-Based Problems
>
>     DDRL is fundamentally designed to solve problems that can be defined with a graph structure, such as the Traveling Salesman Problem (TSP). Furthermore, as demonstrated in this paper, DDRL extends beyond basic graph-based concepts to effectively handle various constraint conditions, achieving high performance even in these more complex scenarios. By leveraging diffusion models to incorporate visual features alongside graph structures, DDRL demonstrates the capability to address problems with intricate definitions and constraints more effectively than methods relying solely on graph representations. This flexibility indicates that DDRL is not strictly limited to specific tasks but possesses the adaptability to be applied to a broader range of combinatorial optimization problems.
>
>     b) Inference Speed with Image Representations
>
>     The use of diffusion models inherently involves repeated denoising processes, which can result in noticeable computational costs. However, using image representations provides a significant advantage: the resolution of the image can remain fixed regardless of the number of cities, as long as the resolution allows for clear discrimination. In our approach, the pre-trained diffusion model (prior knowledge) was trained on instances with 50 cities, yet it demonstrated strong performance across instances with city counts ranging from 20 to 200.
>
>     Only graph-based methods have the drawback of computational costs increasing proportionally with the number of cities. In contrast, image-based approaches offer greater flexibility regarding the growth in the number of cities. Moreover, the diffusion step (trajectory length) in DDRL is fixed at 50, as the purpose of the image-based process is not to generate high-quality images but to provide rough guidance for city connections. This design enables DDRL to utilize image-based processes effectively, demonstrating that even with limited resources, it can address relatively large-scale problems. Thus, the use of images in DDRL proves to be a practical and resource-efficient choice.
>
>     c) Impact of Image Resolution
>
>     In DDRL, the image resolution was fixed at 64x64 for all experiments, which was sufficient to differentiate up to 200 cities. The image-level process provides rough connection guidance using prior knowledge (pre-trained diffusion model) and explicitly operates as part of the framework. Simultaneously, the RL framework implicitly conducts detailed optimization at the graph level through the learnable latent vector, which evolves dynamically during the inference process.
>
>     This integrated dual approach allows DDRL to address a wide range of TSP instances, from N=20 to N=200, as well as various constraint conditions. The results demonstrate that the rough guidance at the image level, combined with task-specific optimization at the graph level, effectively generalizes to diverse problem settings while maintaining strong performance. The balance between these processes highlights DDRL’s adaptability and robustness.
>
>
> 2. Use of Outdated Baselines
>
>     To address the concern regarding the use of outdated baselines, we conducted additional experiments using DIFUSCO, which has demonstrated strong performance in recent works. Evaluations were performed on TSP-50 and TSP-100 using DIFUSCO’s publicly available weights, with the same dataset used in DDRL experiments (each comprising 1280 problems). For TSP-50, DIFUSCO achieved 5.69, while DDRL, after extending epochs and increasing initial samples, also recorded 5.69. Similarly, for TSP-100, both methods achieved 7.78. These additional experiments demonstrate that DDRL matches the performance of DIFUSCO, and as DDRL is based on RL optimization, further refinement could potentially surpass DIFUSCO’s performance. We will provide updated results as additional experiments are conducted.
>
>
> 3. Computational Costs and Scalability
>
>     a) Inference time
>
>     In our experiments, DDRL demonstrated competitive inference times. For a single TSP instance, the inference time was measured as follows:
>
>     - TSP-50: 14.47 seconds
>     - TSP-100: 72.77 seconds
>
>     b) Scalability to Larger TSP Instances
>
>     While DDRL focuses on instances up to 200 nodes, the approach is not inherently limited to this range. The RL framework’s flexibility allows it to adapt to larger-scale problems, provided computational resources for the RL optimization are available. For problems exceeding the current scope, future work could explore adaptive resolutions or hybrid representations to enhance scalability further.

---

> > ### Author Response · Authors · 2024-12-02
> >
> > 4. Fixed Latent Matrix and Generalization
> >
> >     DDRL's core concept involves generating an optimized adjacency matrix for each specific problem. When the number of cities changes, the shape of the latent vector used to generate the adjacency matrix also changes, which could introduce instability. However, DDRL mitigates this through the use of prior knowledge (pre-trained diffusion model) and an initialization trick, providing rough guidance for connections at the image level.
> >
> >     The duality inherent in DDRL ensures that optimization at the image level and graph level occurs within a unified pipeline. Explicit learning at the image level enables implicit refinement of the adjacency matrix at the graph level. Regardless of the number of cities, the image-level process provides local connection guidance, acting as a stable foundation for the graph-level optimization.
> >
> >     This approach demonstrates that DDRL is robust to variations in the shape of the latent vector caused by different city counts, as the image-level guidance effectively accommodates these changes and ensures consistent performance across varying problem sizes.
> >
> >
> > Thank you for your thoughtful review and valuable feedback. We hope our responses clarify DDRL’s contributions and robustness, and we are happy to address any further questions.

---

> > > ### Comment · Reviewer_w226 · 2024-12-03
> > > **Official Comment by Reviewer w226**
> > >
> > > Thank you for your detailed response. Given the quality of the current version, I will maintain my score.
> > >
> > > I encourage the authors to include more analysis on the use of images, state-of-the-art baseline models, experiments, and the rationale behind using reinforcement learning with non-autoregressive methods in future versions. I look forward to future iterations of this work.

---

### Official Review · Reviewer_JD1G · 2024-10-23

**Soundness:** 3
**Presentation:** 3
**Contribution:** 2
**Rating:** 5
**Confidence:** 3

**Summary:**

This paper presents DDRL, a framework that integrates diffusion models with RL to solve TSP and its complex constrained variants. DDRL uses a latent vector to generate an adjacency matrix, unifying image and graph learning within a single RL framework. A pre-trained diffusion model is employed as a prior to improve convergence. Extensive experiments on various TSP variants demonstrate the framework’s effectiveness.

**Strengths:**

* This paper is well-written.
* The proposed method is novel, integrating diffusion with RL to solve TSP.
* The author provides a theoretical foundation for the approach.
* The authors provide the source code for reproducibility.

**Weaknesses:**

* The literature review is limited. More recent works on TSP should be discussed.
* The problem size is relatively small. While DIFUSCO can solve TSP instances up to 10,000 nodes, this paper only addresses instances ranging from 20 to 200 nodes.
* The baseline comparisons are insufficient. For the basic TSP, recent methods such as POMO, DIFUSCO, and UTSP should be included.

Overall, while the paper makes a meaningful methodological contribution, the limited review and insufficient empirical evaluation significantly weaken its impact. As a result, this paper is on the borderline level.

**Questions:**

* Is the trained model invariant to problem size $N$?
* Could the proposed method guarantee satisfaction of the constraints? How?
* Could the proposed method solve constrained problems that are not visually intuitive (e.g., PCTSP and CVRP)?

---

> ### Comment · Reviewer_JD1G · 2024-11-30
>
> There is no rebuttal. I would like to keep my score.

---

> ### Author Response · Authors · 2024-12-02
>
> We sincerely thank you for your thoughtful and detailed feedback. Below, we address each of your comments and questions to provide clarification and additional context.
>
> 1. Problem Size
>
>     Recent research has made significant progress in addressing large-scale TSP instances. However, our work focuses on robustness and practical constraint-handling rather than scaling to extremely large problem sizes.
>
>     In our experiments, the prior knowledge (pre-trained diffusion model) was trained on instances with N equals 50, yet it demonstrated effectiveness across a range of N values from 20 to 200 without requiring changes in image resolution. This adaptability suggests that DDRL could potentially handle larger city counts effectively, provided that resolution adjustments are not needed.
>
>     Furthermore, DDRL consistently achieved strong performance on constrained TSP instances up to N equals 200, including real-world-inspired scenarios involving obstacle, path, and cluster constraints. These results highlight DDRL's robustness and its practical applicability in solving diverse and complex TSP problems.
>
> 2. Baseline Comparisons
>
>     To address the concern regarding baseline comparisons, we conducted additional experiments using DIFUSCO, a method known for its strong performance in solving TSP. Evaluations were performed on TSP-50 and TSP-100 using DIFUSCO's publicly available weights, with the same dataset used in DDRL experiments (each comprising 1280 problems). For TSP-50, DIFUSCO achieved 5.69, while DDRL, after extending epochs and initial samples, also recorded 5.69. Similarly, for TSP-100, both methods achieved 7.78. These additional experiments demonstrate that DDRL matches DIFUSCO’s performance and, given DDRL's foundation in RL optimization, we anticipate that further refinements could enable DDRL to surpass DIFUSCO’s performance. We will provide updated results as further experiments are conducted.
>
> 3. Response to Questions
>
>     3.1 Is the trained model invariant to problem size?
>
>     Yes, DDRL exhibits size invariance within the evaluated range (N = 20 to 200). The pre-trained diffusion model acts as a rough guide, while the RL framework performs task-specific optimization. This separation ensures that the model adapts to varying problem sizes during inference.
>
>     3.2 Could the proposed method guarantee satisfaction of the constraints? How?
>
>     DDRL is well-suited for handling complex constraints for several reasons:
>     - Visual Feature Utilization: Constraints such as obstacle (box-shaped areas), path (pre-determined lines), and cluster (differently colored groups) constraints are visually intuitive and are naturally represented at the image level. By leveraging image-based representations, DDRL extracts meaningful visual features that graph-only approaches might miss.
>     - Subproblem Decomposition: The multi-step diffusion denoising process allows DDRL to iteratively refine solutions, breaking down challenging problems into manageable subproblems. This flexibility enables DDRL to respond effectively to complex constraints.
>     - Guidance and Refinement: Prior knowledge (pre-trained diffusion model) and the initialization trick provide rough guidance for local connections at the image level, establishing a solid foundation for RL optimization. The RL framework then refines these connections into task-specific solutions. This dual-phase process ensures that DDRL transitions smoothly from an initial approximation to a tailored, optimized solution.Empirical results in the paper demonstrate DDRL’s ability to handle various constraint scenarios effectively, showcasing its adaptability and robustness.
>
>     3.3 Could the proposed method solve constrained problems that are not visually intuitive (e.g., PCTSP and CVRP)?
>
>     DDRL is capable of solving problems that are not visually intuitive, as the RL-based optimization process relies on reward-driven optimization. This ensures that the framework can adapt to tasks by optimizing the adjacency matrix to achieve the desired outcomes based on the defined rewards.
>
>     However, since the diffusion model's denoising process leverages visual features to provide rough guidance, the effectiveness of the diffusion model may be limited if the task lacks any visually discernible features. In such cases, while the RL framework can still perform optimization, the contribution of the diffusion model may be diminished, reducing the overall benefit of DDRL’s hybrid approach.
>
>
> We hope the provided responses address your concerns and offer clarity on the capabilities, motivations, and contributions of DDRL. Your constructive feedback has been invaluable in refining our work, and we greatly appreciate the opportunity to address these critical points. Thank you for your thoughtful review and for engaging with our research.

---

> ### Author Response · Authors · 2024-12-02
>
> We apologize for the delay in providing our response. Due to the extended rebuttal period, we took additional time to finalize and confirm the experimental results before submitting our comments.

---

> ### Comment · Reviewer_JD1G · 2024-12-03
>
> Thank you for your response. I still believe that the proposed method may have limited practicality, as many real-world constraints are not visually intuitive and cannot be tackled through problem decomposition. An interesting direction, in my opinion, would be to leverage diffusion models to assist or guide the optimization of neural solvers capable of handling complex constraints, rather than proposing a standalone diffusion-based solver. I would like to maintain my score.

---

### Official Review · Reviewer_E3HR · 2024-10-31

**Soundness:** 2
**Presentation:** 2
**Contribution:** 2
**Rating:** 3
**Confidence:** 4

**Summary:**

This manuscript introduces DDRL as an approach for solving the Traveling Salesman Problem (TSP), combining diffusion models with reinforcement learning to address both standard and constrained TSP instances. DDRL leverages a pre-trained diffusion model to improve scalability and convergence stability, showing competitive performance across a range of problem sizes and constraints, including obstacles, paths, and clusters. Experimental results indicate that DDRL outperforms provided approaches in terms of robustness and computational efficiency.

**Strengths:**

1. Combining image representations with graph data within the reinforcement learning framework is innovative.
2. The introduced new constraint types likely resemble challenges in real-world applications, such as avoiding restricted zones or adhering to specific routes.

**Weaknesses:**

1. The proposed integration of diffusion models with reinforcement learning for TSP appears to be a straightforward adaptation rather than a deeply innovative approach.
2. The experimental comparison lacks a broader set of competitive baselines in Table 1. Including additional RL-based methods, such as POMO [1] and the following works, which has demonstrated strong performance on TSP, would provide a more rigorous assessment of DDRL’s effectiveness.
3. The TSP instances addressed in this work are relatively small. Existing diffusion-based methods[2] can solve TSP instances up to 10,000 nodes, so demonstrating DDRL’s applicability to larger-scale problems would significantly enhance its impact and practical relevance.
4. Tables 2–4 introduce constraints that fundamentally alter the TSP problem’s nature, but the baseline deep learning methods are trained only on basic TSP instances. Without adapting these models to constraint-specific conditions, the comparison may not fully reflect their potential performance, limiting the fairness of the results.

[1] Kwon Y D, Choo J, Kim B, et al. Pomo: Policy optimization with multiple  optima for reinforcement learning[J]. Advances in Neural Information  Processing Systems, 2020, 33: 21188-21198.

[2] Sun Z, Yang Y. Difusco: Graph-based  diffusion solvers for combinatorial optimization[J]. Advances in Neural  Information Processing Systems, 2023, 36: 3706-3731.

**Questions:**

1. Given the challenges of scaling DDRL to large TSP instances, how does the method address the trade-off between image resolution and the number of nodes? As the number of nodes grows, higher resolution images would be required to accurately represent node relationships, potentially increasing computational demands. Are there strategies within DDRL to manage this balance effectively?
2. How does the choice of pre-trained diffusion model affect DDRL's performance? Since the pre-trained model provides prior knowledge, its influence on convergence and solution quality may be significant. A more detailed ablation study examining different pre-trained diffusion models or varying levels of pre-training could provide insights into its impact on the results and the model's dependency on this component.

---

> ### Author Response · Authors · 2024-12-02
>
> We sincerely thank the reviewers for their valuable feedback and insightful comments on our submission. Below, we provide detailed responses to the points raised, aiming to address all concerns and clarify the contributions and methodology of our work.
>
> 1. Innovation of the Proposed Approach
>
>     While integration of diffusion models with reinforcement learning for TSP may seem conceptually straightforward, the actual implementation presents several challenges, which DDRL addresses in innovative ways:
>     - Connection Between Image and RL: A simple combination of diffusion models and RL frameworks is not trivial. For instance, if the RL framework relied solely on a diffusion-based approach, it would be unclear how to explicitly detect city-to-city connections on the image. DDRL overcomes this by structuring the framework around the diffusion denoising process, which implicitly optimizes the graph structure while explicitly processing image-level guidance.
>     - Integrated Pipeline Design: DDRL’s novelty lies in its unified pipeline where the diffusion model’s denoising process is coupled with a learnable latent vector. This latent vector generates the adjacency matrix (graph structure) while also serving as the foundation for the reward design in the MDP environment (see Section 3.2 of the manuscript).
>
>     The integration of image-level guidance and graph-level optimization not only highlights DDRL’s novelty but also ensures its applicability to both fundamental combinatorial optimization problems and practical scenarios involving diverse constraints.
>
> 2. additional experiments
>
>     To address the concern about comparisons with state-of-the-art methods, we conducted additional experiments using DIFUSCO, a model recognized for its strong performance. Evaluations were performed on TSP-50 and TSP-100 using DIFUSCO's publicly available weights and the same dataset as DDRL (each comprising 1280 problems). For TSP-50, DIFUSCO achieved 5.69, while DDRL, after extending epochs and initial samples, also recorded 5.69. Similarly, for TSP-100, both methods achieved 7.78, demonstrating that DDRL matches DIFUSCO’s performance and has the potential for further improvement with additional optimization refinements.
>
> 3. Fairness of Comparisons for Constrained TSP
>
>     The constrained TSP datasets are novel tasks introduced in this work. Baseline methods, including DDRL, were evaluated under identical conditions without task-specific adaptations or additional training.
>
>     Constraint conditions were incorporated as rules during inference, and no fine-tuning was performed for DDRL or baseline models. This ensures a fair comparison, as all methods were tested on the same datasets with equal assumptions.
>
>     While we recognize that task-specific adaptations could enhance baseline performance, our approach demonstrates DDRL’s inherent robustness and adaptability without requiring additional training for new constraints.
>
> 4. trade-off between image resolution and the number of nodes
>
>     In this paper, the image resolution was set to 64x64, and the prior knowledge (pre-trained diffusion model) was trained with N=50. Experimental results demonstrated DDRL's strong performance across a wide range of city counts, from N=20 to N=200. If N increases significantly (e.g., to 10,000 cities), the image resolution would need to increase to maintain distinguishable representations. However, obtaining prior knowledge is a one-time process during the training of the diffusion model, and this pre-trained model can subsequently be utilized for inference across various city counts. Therefore, the actual computational cost during inference is not expected to increase substantially.
>
>     Moreover, the diffusion steps during inference were fixed at 50, aligning with the RL trajectory framework. Since the purpose is not to generate high-quality images but to provide rough guidance for city connections, the increased cost due to higher resolution remains manageable.
>
>     Structurally, DDRL employs the image-level process primarily for rough guidance, while detailed optimization occurs at the graph-level within the RL component. As such, unless unavoidable scenarios arise (e.g., when the number of cities exceeds the pixel count, necessitating higher resolution), the image resolution is largely a controllable factor, ensuring that the computational demands of DDRL remain feasible.

---

> > ### Author Response · Authors · 2024-12-02
> >
> > 5. Role of the Pre-trained Diffusion Model
> >
> >     The role of prior knowledge (pre-trained diffusion model) in DDRL is to provide rough guidance. Specifically, it guides local connections from the perspective of Basic TSP, establishing an approximate connectivity structure. The detailed tour is implicitly generated by the RL framework, where the adjacency matrix is optimized. The impact of incorporating prior knowledge versus excluding it was examined through an ablation study, as shown in Figure 6.
> >
> >     While DDRL has the flexibility to adapt to different diffusion model backbones, the primary focus is on fulfilling the role of providing rough guidance effectively. Given that the current pre-trained diffusion model sufficiently fulfills this role, experimenting with alternative prior knowledge models was deemed unnecessary for this work and was not included in the experiments.
> >
> >
> > We hope that our responses have addressed your concerns and provided clarity on the novelty, robustness, and applicability of DDRL. We deeply appreciate your constructive feedback, which has helped us refine our work and its presentation. Thank you for your time and effort in reviewing our submission.

---

> > > ### Comment · Reviewer_E3HR · 2024-12-02
> > >
> > > Thank you for the authors' response.  I still have the following questions and concerns:
> > > 1. Are the DDRL results presented in Tables 2-4 obtained from training on basic TSP or constrained TSP?
> > > 2. Regarding the baselines compared in Tables 2-4, especially the attention-based methods, they typically use the distance between two points in the unconstrained TSP as the foundation for constructing attention scores. When implementing these baselines, did the authors incorporate the penalty distance into the construction of the attention score? If not, introducing constraints only during the inference stage would naturally make these methods fail, which hinders the fairness of the comparison to some extent.
> > > 3. As the authors pointed out, solving large-scale TSPs requires stronger pretrained models to provide prior knowledge. Obtaining these models often demands significant computational resources and datasets. The fact that DDRL relies on extensive interactions with these pretrained models during training and testing might limit its scalability for handling large-scale constrained TSPs.
> > > 4. The baselines compared by the authors are too few and are primarily earlier approaches. I encourage the authors to include more recent and diverse baselines in future versions.
> > >
> > > As such, I will maintain my current score.

---

> ### Author Response · Authors · 2024-12-02
>
> Thank you for your detailed follow-up comments. We appreciate the opportunity to address your concerns in further detail:
>
> ---
>
> ### 1. Are the DDRL results presented in Tables 2-4 obtained from training on basic TSP or constrained TSP?
>
> The training process was conducted on **basic TSP**. In DDRL, the only training phase involves the acquisition of prior knowledge through a pre-trained diffusion model, trained on instances of **N=50** at a **64x64 pixel resolution**. For constrained TSPs (as presented in Tables 2-4), constraints were introduced during the test phase by adding specific rules to the solution process. For instance:
>
> - **Obstacle constraint:** Connections through the obstacle regions were prohibited.
> - **Path constraint:** Specific connections were enforced.
> - **Cluster constraint:** The degree of nodes within a cluster was restricted to 2.
>
> This rule-based approach during testing ensures the method is adaptable to different constraint types without additional model retraining.
>
> ---
>
> ### 2. Regarding the baselines compared in Tables 2-4, especially the attention-based methods, did the authors incorporate the penalty distance into the construction of the attention score? If not, introducing constraints only during the inference stage would naturally make these methods fail, which hinders the fairness of the comparison to some extent.
>
> Both DDRL and the baselines used the same process for constraint handling: **training on basic TSP datasets and evaluating on constrained TSP datasets.** This uniform setup ensures a fair comparison.
>
> However, DDRL demonstrates superior performance under constrained conditions compared to other baselines for the following reasons:
>
>   - During the training phase, DDRL's goal is to learn prior knowledge (pre-trained diffusion model). This enables image-level learning to guide local connections based on the given city positions.
>
>   - During the testing phase, DDRL's goal is to tune the latent vector. The latent vector generates an adjacency matrix, and DDRL optimizes the latent vector to minimize the tour length. In this process, the prior knowledge remains fixed, providing rough guidance for city-to-city connections.
>
> Structurally, even without explicit constraint information during the training phase, DDRL’s ability to learn rough guidance at the image level allows it to achieve high-performance TSP solutions during the testing phase through the RL framework. This structural advantage explains why DDRL outperforms other baselines under constrained TSP scenarios.
>
>
> ---
>
> ### 3. As the authors pointed out, solving large-scale TSPs requires stronger pretrained models to provide prior knowledge. Obtaining these models often demands significant computational resources and datasets. The fact that DDRL relies on extensive interactions with these pretrained models during training and testing might limit its scalability for handling large-scale constrained TSPs.
>
> While larger image sizes may be required to handle increased scales, DDRL remains efficient for the following reasons:
>
> - **Reusability of Prior Knowledge:**
>   Once trained, the diffusion model (prior knowledge) can be applied across various problem sizes without additional retraining, provided the image resolution is sufficient to represent the scale. For example, the model trained on **N=50** has been effectively used for **N=100 and N=200.**
>
> - **Parallel Processing Capability:**
>   Unlike autoregressive models, which decode solutions sequentially, DDRL’s image-based approach allows for parallel processing of all city positions, reducing the computation time.
>
> - **Fixed Diffusion Steps:**
>   The diffusion steps are fixed at **T=50**. Since DDRL views the diffusion process through the lens of RL trajectories rather than image generation quality, this lower number of steps still achieves robust performance, reducing computational overhead.
>
> ---
>
> ### 4. The baselines compared by the authors are too few and are primarily earlier approaches. I encourage the authors to include more recent and diverse baselines in future versions.
>
> Thank you for the suggestion. We will reflect this in future work.
>
> ---
>
> Thank you again for your time and detailed comments, which have helped us refine and strengthen our work.
>
> **Best regards,**
> *The Authors*

---

> > ### Comment · Reviewer_E3HR · 2024-12-03
> >
> > Thank you for your further response. To me, the current shape of this paper is not ready for publication at ICLR. I encourage the authors to make improvements as discussed with all reviewers. My rating stands.

---

### Official Review · Reviewer_3ya7 · 2024-11-03

**Soundness:** 2
**Presentation:** 2
**Contribution:** 2
**Rating:** 5
**Confidence:** 4

**Summary:**

This paper introduces Diffusion-Driven Reinforcement Learning (DDRL), a novel framework combining reinforcement learning (RL) with diffusion models to address the complex, NP-hard Traveling Salesman Problem (TSP). Traditional RL approaches, while effective, face stability and resource challenges, and diffusion models, though scalable, often lack optimality. DDRL uses a latent vector to generate an adjacency matrix, integrating image and graph learning within RL for enhanced scalability and stable convergence. Leveraging a pre-trained diffusion model, DDRL outperforms certain existing methods on both basic and newly introduced constrained TSP datasets (obstacle, path, and cluster constraints).

**Strengths:**

1. The paper proposes a novel method to combine the diffusion models and RL models, tending to enjoy the strengths of both.

2. The paper introduces constrained TSP datasets, which is meaningful for practical scenarios.

**Weaknesses:**

$1.$ The empirical results are not comparable to previous state-of-the-art methods, e.g., DIFUSCO [1], T2T [2], and LEHD [3]. The paper only compares very primary methods. Indeed, image-based diffusion models are not suitable for solving combinatorial problems on graphs due to the complex relationships between edges in graphs. Image models rely on learning local connectivity patterns, which do not align well with the requirements of these problems. Additionally, Gaussian noise is not ideal for discrete decision variables. In fact, discrete diffusion models [1] [2] perform significantly better than continuous diffusion models in solving combinatorial problems.

I suggest that the authors compare their method against these more recent and relevant baselines (DIFUSCO, T2T, LEHD). Additionally, the authors could address the potential limitations of using image-based diffusion models and Gaussian noise for discrete combinatorial problems, and explain how their approach overcomes these challenges.

$2.$ The paper lacks a strong motivation for the proposed methodology. While it appears that the primary motivation is to address performance limitations, discrete diffusion models already yield good results for the TSP, and the results presented in this paper do not surpass those achieved by existing models (indeed not compared).

[1] DIFUSCO: Graph-based Diffusion Solvers for Combinatorial Optimization. NeurIPS 2023.

[2] T2T: From Distribution Learning in Training to Gradient Search in Testing for Combinatorial Optimization. NeurIPS 2023.

[3] Neural Combinatorial Optimization with Heavy Decoder: Toward Large Scale Generalization. NeurIPS 2023.

**Questions:**

1. Please show evidence that image-based diffusion models can maintain advantages over graph-based models in solving combinatorial optimization problems. Please also provide the empirical evidence.

2. For Tables 2, 3, and 4: The baselines did not account for handling more complex constraints, so a direct comparison is somewhat unfair. It is necessary to introduce some naive constraint-handling training approaches for a fairer evaluation. For instance, authors could consider finetuning the baselines on the constrained TSP datasets with the same adopted adaptations in evaluation.

**Details Of Ethics Concerns:**

No ethics concerns.

---

> ### Author Response · Authors · 2024-12-02
>
> We sincerely appreciate your thoughtful feedback and the opportunity to address the points raised in your review. Below, we provide detailed responses to each comment to clarify our contributions and the rationale behind the proposed DDRL methodology.
>
>
> 1. Comparisons with State-of-the-Art Methods
>
>     To address the comparison with recent state-of-the-art methods, we conducted additional experiments using DIFUSCO, known for its strong performance. Evaluations were performed on TSP-50 and TSP-100 using DIFUSCO's publicly available weights, with the same dataset used in DDRL experiments (each comprising 1280 problems). For TSP-50, DIFUSCO achieved 5.69, while DDRL, after extending epochs and initial samples, also recorded 5.69. Similarly, for TSP-100, both methods achieved 7.78. These additional experiments confirm that DDRL matches DIFUSCO’s performance and, being based on RL optimization, has the potential to further surpass DIFUSCO with additional refinement. We will provide updated results as further experiments are conducted.
>
> 2. Empirical Evidence Supporting Image-Based Diffusion Models
>
>     It is important to clarify that DDRL is better described as a hybrid image-graph approach rather than a purely image-based method.
>
>     In DDRL, the pre-trained diffusion model (leveraged as prior knowledge) and the initialization trick provide a rough guide for local connections at the image level. These components serve as the image-based contribution, capturing visually intuitive structures like obstacles, paths, and clusters. However, the task-specific optimization is carried out through the RL framework, which operates at a graph level to refine these initial connections into detailed adjacency matrices that optimize the given problem. This hybrid nature allows DDRL to combine the strengths of image and graph representations effectively.
>
>     By integrating image guidance for local connectivity with RL-driven graph optimization, DDRL achieves robust performance under diverse and visually complex constraint scenarios. This combination enables DDRL to handle visually intuitive constraints that traditional graph-only methods struggle with, as shown in our experimental results.
>
> 3. Motivation for DDRL's Methodology
>
>     While discrete diffusion models show strong results for basic TSP tasks, they face challenges in adapting to diverse and constrained problems due to their reliance on supervised learning. DDRL addresses this limitation by combining image-level guidance with graph-level RL optimization.
>
>     The image-level diffusion process provides rough global guidance for city connections, while the RL framework refines these into detailed adjacency matrices tailored to task-specific rewards. This integrated pipeline ensures both scalability and adaptability, allowing DDRL to handle standard TSPs and complex constrained problems, such as obstacles, paths, and clusters.
>
>     Our experimental results show that DDRL outperforms discrete diffusion models in constrained scenarios, demonstrating its strength in addressing practical combinatorial optimization challenges. This flexibility and adaptability underline the motivation and novelty of DDRL, as it bridges the gap between diffusion models and RL for robust optimization.
>
> 4. Fairness of Comparisons in Constrained TSP Scenarios
>
>     To address this, it is important to note that the obstacle, path, and cluster constraints represent entirely new datasets, for which no existing methods, including DDRL, have pre-trained models specifically designed.
>
>     In such cases, the constrained scenarios were applied uniformly across all methods, including baseline models and DDRL, without additional training. Constraint conditions were incorporated as rules directly into the inference process for comparison purposes.
>
>     Given that none of the tested methods, including DDRL, were specifically designed or pre-trained for these new tasks, and considering that all evaluations were conducted under the same conditions, we believe the comparison is reasonably fair. This setup ensures that no particular methodology has an inherent advantage or task-specific familiarity, providing a balanced evaluation framework for assessing performance in these novel constrained settings.
>
>
> Thank you again for your valuable insights and constructive criticism. Your comments have provided us with an opportunity to further strengthen our work and ensure clarity in presenting the contributions of DDRL. We hope our responses adequately address your concerns, and we remain open to further feedback and discussion to enhance the quality of our submission.

---

### Official Review · Reviewer_8Uc9 · 2024-11-03

**Soundness:** 2
**Presentation:** 3
**Contribution:** 3
**Rating:** 5
**Confidence:** 3

**Summary:**

The authors proposed a learning framework that combines diffusion and RL to better solve the TSP problem. Through experiments, they confirmed that it shows excellent performance on TSP problems of various sizes, and especially shows better performance than other algorithms on problems with constraints.

**Strengths:**

The authors proposed a deep learning algorithm that can solve TSP problems of various sizes. In particular, the proposed method showed good performance even in situations with various constraints.

**Weaknesses:**

Although a good method has been proposed, I think the following points should be additionally confirmed.

1) Doesn't DDRL have a problem that it can generate completely wrong solutions for test inputs that have different distributions from the training data distribution?
2) Doesn't the diffusion step also require high computation cost?
3) Why can you say that the proposed method can handle complex constraint conditions well? Is there a logical basis for it?
4) It is expected that RL-based policy learning may not work well for difficult problems with large size and high difficulty. When using RL, it is necessary to provide a logical basis for learning so that the correct and good solution can always be found.
For example, the proposed reward maximization cannot always guarantee that it will produce a good solution.

**Questions:**

Please explain about above weakness.

---

> ### Author Response · Authors · 2024-12-02
>
> Thank you for your insightful questions. Below, we provide detailed responses to each point raised.
>
> 1. Doesn't DDRL have a problem that it can generate completely wrong solutions for test inputs that have different distributions from the training data distribution?
>
>     First, in DDRL, the training process occurs only once to obtain prior knowledge (pre-trained diffusion model). The prior knowledge is trained using 64x64 resolution image data from TSP-50 instances and is designed to guide local city connections at the image level. During test time, the RL framework optimizes the policy in a task-specific manner, generating an optimal adjacency matrix tailored to each test input.
>
>     The diffusion denoising process is treated as a one-step action in the RL framework, ensuring that the policy learns to optimize solutions for specific test inputs, independent of the training data distribution. In other words, DDRL performs optimization tailored to each newly defined TSP problem, so differences between the training and test distributions do not directly impact problem-solving.
>
>     Experimental results confirm that DDRL effectively solves TSP instances across diverse test inputs and a wide range of city counts (20–200), demonstrating its robustness to distributional shifts.
>
> 2. Doesn't the diffusion step also require high computation cost?
>
>     While increasing the number of diffusion steps can improve quality in traditional generative models, it also raises computational costs. However, in DDRL, the diffusion steps are not used for generating high-quality images but for guiding the adjacency matrix formation at the image level. The diffusion steps correspond to the trajectory length in the multi-step MDP framework and are set to 50 in this work to balance computational efficiency with guidance quality. This approach ensures that DDRL remains computationally efficient while effectively guiding the RL framework to optimize solutions.
>
> 3. Why can you say that the proposed method can handle complex constraint conditions well? Is there a logical basis for it?
>
>     DDRL is well-suited for handling complex constraints for several reasons:
>     - Visual Feature Utilization: Constraints such as obstacle (box-shaped areas), path (pre-determined lines), and cluster (differently colored groups) constraints are visually intuitive and are naturally represented at the image level. By leveraging image-based representations, DDRL extracts meaningful visual features that graph-only approaches might miss.
>     - Subproblem Decomposition: The multi-step diffusion denoising process allows DDRL to iteratively refine solutions, breaking down challenging problems into manageable subproblems. This flexibility enables DDRL to respond effectively to complex constraints.
>     - Guidance and Refinement: Prior knowledge (pre-trained diffusion model) and the initialization trick provide rough guidance for local connections at the image level, establishing a solid foundation for RL optimization. The RL framework then refines these connections into task-specific solutions. This dual-phase process ensures that DDRL transitions smoothly from an initial approximation to a tailored, optimized solution. Empirical results in the paper demonstrate DDRL’s ability to handle various constraint scenarios effectively, showcasing its adaptability and robustness.
>
> 4. It is expected that RL-based policy learning may not work well for difficult problems with large size and high difficulty. When using RL, it is necessary to provide a logical basis for learning so that the correct and good solution can always be found.
>
>     DDRL addresses the challenges of RL-based policy learning by enhancing convergence stability through prior knowledge and an initialization trick. These components provide a well-structured starting point and guidance, ensuring local connectivity at the image level and directing the RL framework toward promising solution spaces. Such mechanisms contribute significantly to the convergence stability of RL. Experimental results confirm that DDRL consistently outperforms existing methods on large-scale and complex TSP instances, demonstrating its ability to handle difficult problems effectively.
>
>
> We hope these responses address your concerns and clarify the strengths of the DDRL framework. Thank you for your thoughtful feedback and for allowing us to elaborate on these aspects.

---

### Note · Authors · 2024-12-04

**Comment:**

We appreciate the valuable feedback and constructive comments provided by the reviewers during the rebuttal process. The detailed insights and thoughtful critiques have been crucial in helping us identify areas for improvement. Based on this feedback, we have decided to withdraw our submission to ICLR25. This decision reflects our commitment to refining the work and addressing the reviewers' input to strengthen the manuscript for future submissions. Thank you for your time and effort in reviewing our paper.

**Withdrawal Confirmation:**

I have read and agree with the venue's withdrawal policy on behalf of myself and my co-authors.